# Machine learning-based tsunami inundation prediction derived from offshore observations

Iyan E. Mulia [1,2] ✉, Naonori Ueda[1,2], Takemasa Miyoshi[1,3], Aditya Riadi Gusman [4] & Kenji Satake [5]

The world's largest and densest tsunami observing system gives us the leverage to develop a method for a real-time tsunami inundation prediction based on machine learning. Our method utilizes 150 offshore stations encompassing the Japan Trench to simultaneously predict tsunami inundation at seven coastal cities stretching ~100 km along the southern Sanriku coast. We trained the model using 3093 hypothetical tsunami scenarios from the megathrust ($M$w 8.0–9.1) and nearby outer-rise ($M$w 7.0–8.7) earthquakes. Then, the model was tested against 480 unseen scenarios and three near-field historical tsunami events. The proposed machine learning-based model can achieve comparable accuracy to the physics-based model with ~99% computational cost reduction, thus facilitates a rapid prediction and an efficient uncertainty quantification. Additionally, the direct use of offshore observations can increase the forecast lead time and eliminate the uncertainties typically associated with a tsunami source estimate required by the conventional modeling approach.

Motivated by the devastating tsunami of the 2011 Tohoku-oki event, the National Research Institute for Earth Science and Disaster Resilience of Japan installed a large-scale cabled seafloor observatory named the Seafloor Observation Network for Earthquakes and Tsunamis along the Japan Trench (S-net). S-net consists of ocean bottom pressure sensors and seismometers capable of transmitting high-frequency data of 1 kHz in real time[1,2]. Such sophisticated devices would greatly improve the disaster prevention measure, especially for earthquakes and tsunamis originating from the Japan Trench subduction zone and its vicinity. From the tsunami hazard standpoint, the vast coverage and high spatial density of S-net facilitate early detection of earthquakes and tsunamis in the region, and thus in turn leads to effective warnings. S-net has hitherto successfully detected even small tsunamis generated by the 20 August 2016 ($M$w 6.0) off Sanriku[3], the 22 November 2016 ($M$w 7.0) off Fukushima[4], and a far-field event of the 8 September 2017 ($M$w 8.1) Chiapas, Mexico[2] earthquakes.

In addition to the early tsunami detection, an effective warning should incorporate more information, including the potential inundation extent and height pattern at locations of interest. However, the currently operational tsunami early warning systems in the world still do not have a robust system capable of forecasting the tsunami inundation, particularly in the near field. Traditionally, tsunami inundation prediction can be approximated by numerically solving the nonlinear shallow water equations simulating tsunami hydrodynamics[5]. Previous studies showed that this kind of physics-based model could simulate tsunami inundation processes accurately[6,7]. However, the method requires accounting for all the variables in the governing equations, i.e., elevation and velocity components, that evolve in time at all numerical grids.

Consequently, the high computational complexity has been the major obstacle to its real-time application unless a high-performance computing cluster is employed[8,9]. For instance, during the 2011

[1]Prediction Science Laboratory, RIKEN Cluster for Pioneering Research, Kobe, Japan. [2]Disaster Resilience Science Team, RIKEN Center for Advanced Intelligence Project, Tokyo, Japan. [3]Data Assimilation Research Team, RIKEN Center for Computational Science, Kobe, Japan. [4]GNS Science, Lower Hutt, New Zealand. [5]Earthquake Research Institute, The University of Tokyo, Tokyo, Japan. ✉e-mail: iyan.mulia@riken.jp

Tohoku-oki event, the primary crest of the tsunami arrived ~44 min after the earthquake in our study area[10]. An inundation prediction using the conventional physics-based model on a standard computer takes approximately 30 min, and an additional time of ~5–35 min is needed to estimate the source depending on the dataset[11]. Furthermore, the conventional method relies on the tsunami source estimate quality, often associated with large uncertainties[12]. Therefore, a method that can directly extract the characteristic of tsunamis offshore, without using the tsunami source estimate, as the basis to build an inundation prediction model is desired. To that end, we apply a machine learning technique based on artificial neural networks that is expected to exploit the full potential of S-net in a real-time tsunami prediction.

As a typical data-driven approach, a prediction model by neural networks is constructed merely upon statistical correlations between input and output data. In our case, the input consists of tsunami data observed at 150 S-net stations, and the output is the corresponding inundation map at the specified locations. Owing to the dense observation network of S-net, the prediction of maximum tsunami inundation heights (the height above the Mean Sea Level) or flow depths (the height above the ground) and coverage by neural networks can be derived from offshore observations without calculating the entire state variables of the tsunami dynamical system. Therefore, it would substantially reduce the real-time computational cost compared with the physics-based simulation.

Since tsunami occurrence is infrequent, our neural networks model is trained on a large number of precalculated physics-based model results, as explained in the Method section. This strategy in machine learning-based tsunami inundation predictions has been proposed in Fauzi and Mizutani[13] and Mulia et al.[14]. However, due to limited observation points, their methods still require estimating the tsunami source and numerically solving the linearized shallow water equations. Therefore, their machine learning models were designed to transform the linear simulation results into the nonlinear high-resolution inundation map. More recent studies[15,16] share similar means to ours. They utilized offshore tsunami data to predict a series of water level fluctuations at a limited number of land or sea sites, giving detailed information on the tsunami temporal variation at the specified locations. Their models can also be expanded to account for more locations. However, incorporating both temporal and spatial

variations in great detail (e.g., >200,000 target points in our case) is excessive and would eventually increase the computing time and storage.

For near-field cases, spatial information on the maximum tsunami heights or flow depths is needed to inform the evacuation efforts that should start immediately following the detection of tsunamigenic earthquakes. A rapid inundation map could be used to enhance the available map for tsunami evacuation routes and safe zones. It will better inform the authorities involved (e.g., police, firefighter, and operations manager) in executing tsunami evacuation plans. The efficacy of a real-time tsunami inundation map in guiding evacuees during a tsunami evacuation drill had been attested[17]. The inundation map is intuitively understandable to the public as it can be overlaid with geographical maps and visualized via a common online map service, such as Google Maps[17]. Furthermore, cloud computing technology for online tsunami simulations has been proposed[18]. The on-demand computing environment is suitable and cost-effective for a real-time tsunami prediction occasionally accessed only during the event. For our case, the fully trained neural network parameters can be stored within a cloud instance and executed in real time with the given inputs. The online modeling framework can also be configured to facilitate warning dissemination to potential users.

In this study, we design our model to predict the spatial distribution of tsunami inundation flow depths over a considerably broad coastal region with a ~30-m grid resolution (see Fig. 1). We consider seven cities on the southern Sanriku coast: Sanriku, Ofunato, Rikuzentakata, Kesennuma, Motoyoshi, Minamisanriku, and Oppa, as our locations of interest, being one of the most vulnerable regions to tsunamis along the Pacific coast of Japan. The direct use of offshore tsunami observations and the single predictive model for all designated cities are two prominent aspects of our method. These features are an advancement from the existing studies, commonly constructed one model for one city or location for various reasons, mainly computational efficiency[11,13,14]. Incorporating multiple localized domains that are computationally intensive parts in the conventional model to simulate tsunami inundation can be a hassle in a real-time computation because it likely results in an exponentially increased computing time. Whereas a large neural network output vector−a flattened two-dimensional inundation grid into a one-dimensional array−affects only the training time but not necessarily the prediction time, which in

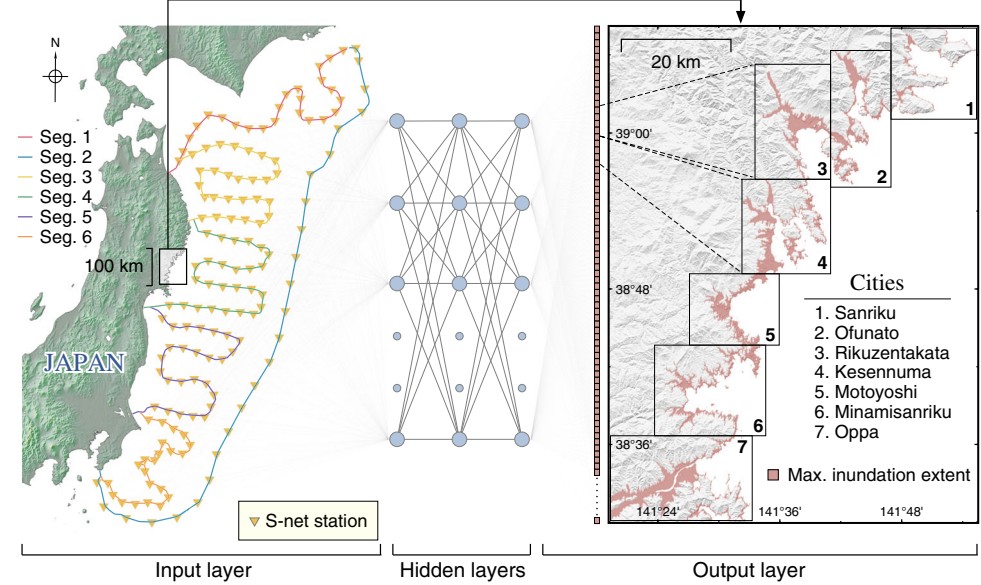

**Fig. 1 | The schematic of the proposed method.** Locations of interest are shown with maximum inundation extent on the training set. S-net station segments are marked with colored lines.

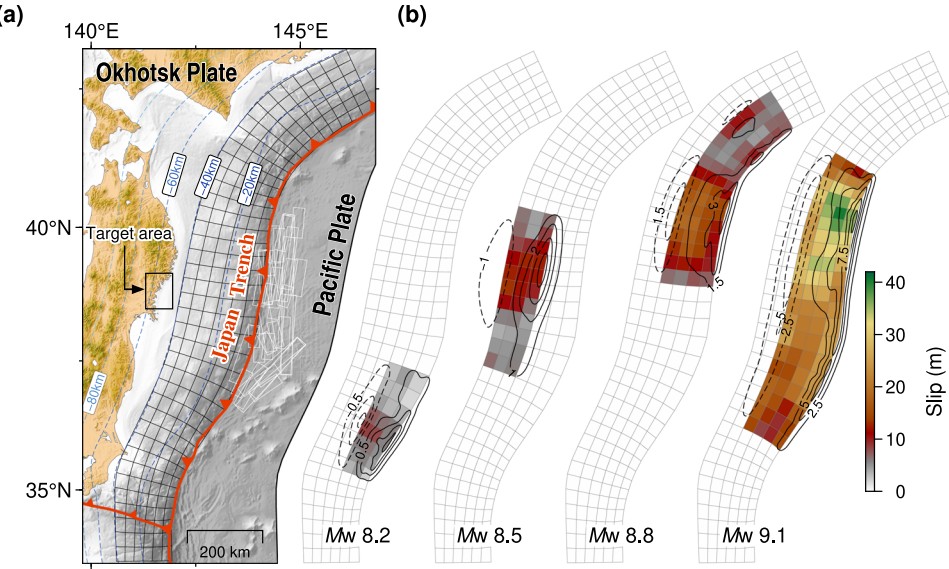

**Fig. 2 | Hypothetical tsunami source scenarios. a** Discretization of the Japan Trench plate interface and outer-rise faults (white rectangles). Dashed contours depict the slab depth[20]. **b** Examples of stochastically generated slip on the megathrust fault. The dashed and solid black contours indicate the coseismic subsidence and uplift, respectively, with intervals of 0.5 m for $M$w 8.2, 1 m for $M$w 8.5, 1.5 m for $M$w 8.8, and 2.5 m for $M$w 9.1.

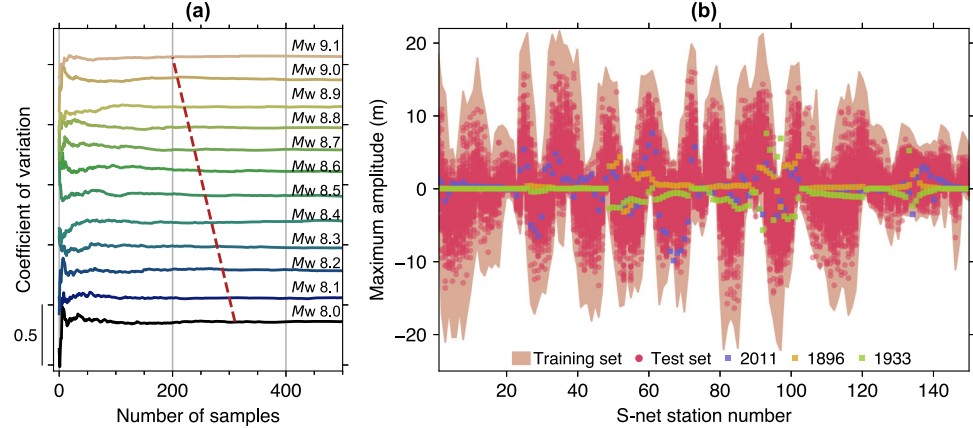

**Fig. 3 | Optimal number of scenarios and dataset range. a** Coefficient of variation of maximum tsunami amplitudes within 20 min at S-net stations for all considered earthquake magnitudes. The red dashed line indicates the requisite number of samples at each magnitude. **b** Ranges of maximum tsunami amplitude used as inputs in the training and test sets in conjunction with inputs for three real tsunami events.

our case takes only a fraction of a second using a standard computer. The overall schematic procedure of the proposed method is shown in Fig. 1.

## Results

### Tsunami source scenarios

The main contributing factor to the tsunami hazard in the study area is submarine earthquakes generated within the Japan Trench subduction zone (Fig. 2a). For megathrust events, we use a stochastic slip model[19] that randomly generates earthquake slip patterns over a discretized plate interface obtained from the SLAB 2 model[20] (see Method section for details). Examples of the resulting slip for certain earthquake magnitudes are depicted in Fig. 2b. The figure also shows the coseismic displacement from the slip calculated using elastic dislocation theory in a half-space[21]. We then use the calculated sea surface displacement to initiate the tsunami simulation by assuming an instantaneous deformation and a long wave approximation.

The other important tsunami source is the outer-rise earthquake (Fig. 2a). For example, the 1933 Showa Sanriku outer-rise earthquake

caused tsunami damage in the same regions affected by the 1896 Meiji Sanriku earthquake on the plate interface. Baba et al.[22] demonstrated that the maximum tsunami height caused by the largest known outer-rise earthquake ($M$w 8.7) in our study area could reach up to 27 m. The tsunami source scenarios from outer-rise earthquakes are based on fault parameters provided by a previous study[22]. These faults were identified through marine seismic observations and surveys[23–25]. The coseismic displacements are calculated similarly to the megathrust earthquakes but with a uniform slip assumption because the contribution of heterogeneous slips on the identified outer-rise faults to the tsunami variability is only 10–15%[22]. An example of vertical displacement from one of the outer-rise earthquakes considered in this study is shown in Supplementary Fig. 1.

To determine the appropriate number of scenarios for the megathrust earthquakes, we first apply Green's functions summation technique[26,27] described in the Method section. With this technique, we can efficiently generate tsunami waveforms at S-net stations from many earthquake slip scenarios before simulating the inundation. Therefore, this step is independent of the actual simulation of tsunami

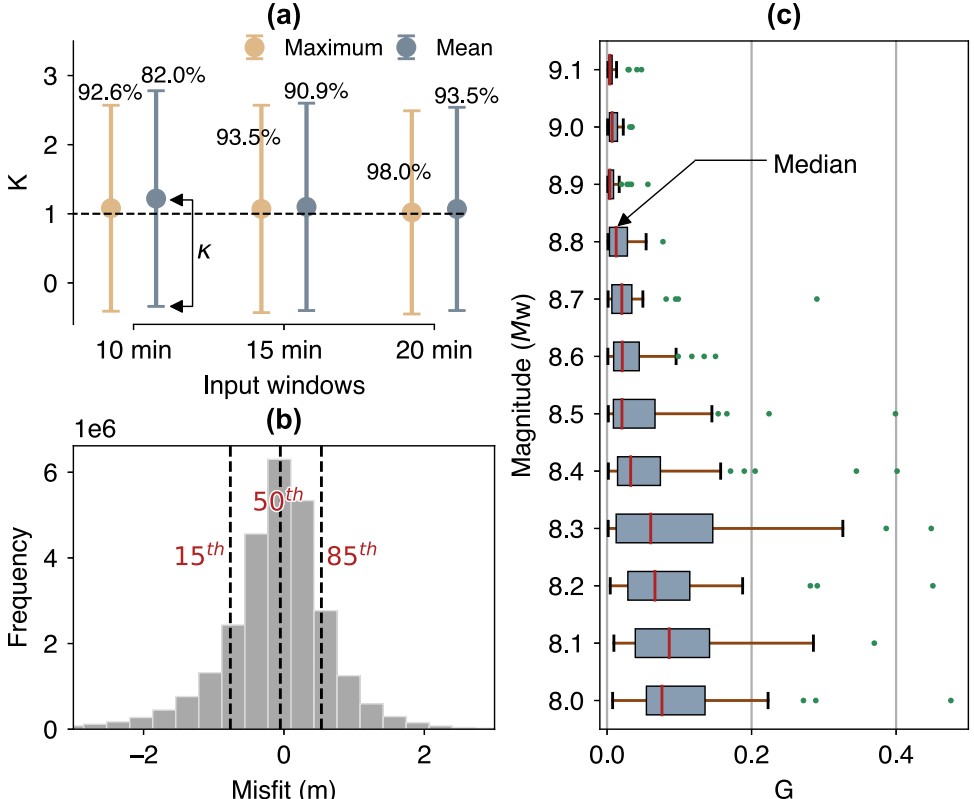

**Fig. 4 | Statistical evaluations of the test set. a** Aida's numbers ($K$ and $\kappa$) and accuracy measure in percentage using maximum and mean tsunami amplitudes as inputs at three prediction time windows of 10, 15, and 20 min. **b** Distribution of misfits with 15th, 50th, and 85th percentiles (dashed lines). **c** Box plot of the goodness-of-fit ($G$) for all samples at each specified earthquake magnitude. **b** and **c** are based on the predictions using the 20-min window of maximum tsunami amplitude.

scenarios used for the training and testing sets. The analysis aims at estimating the degree of variability of inputs concerning the number of samples for various earthquake magnitudes based on a coefficient of variation. Here we consider maximum tsunami amplitudes of bottom pressure sensors at S-net as our machine learning model input. Figure 3a shows the coefficient of variation of model input using 20-min tsunami data at every increment of samples from 2 to 500. The figure illustrates that earthquakes with smaller magnitudes require more samples to reach the convergence state, a condition where the coefficient of variation curve is stable. This is an expected outcome because the spatial variability of locations of smaller earthquakes is greater than the larger ones[27]. Adding more samples beyond the estimated convergence state will no longer contribute to the variability of input, thus is redundant.

According to the above analysis and a visual inspection, the requisite number of scenarios for the smallest ($M$w 8.0) and largest ($M$w 9.1) considered megathrust earthquake magnitudes are 310 and 200, respectively. We then apply linear interpolation to the remaining magnitudes illustrated by the red dashed line in Fig. 3a, resulting in 3060 scenarios. This implies that about 3000 plate interface earthquake scenarios are needed for the training set. The number of identified outer-rise faults is 33, hence the total number of scenarios for the training set is 3093. A list of fault parameters for the outer-rise earthquakes used in this study can be found in Baba et al.[22]. For the test set, we generate 40 hypothetical megathrust earthquakes for each magnitude (12 magnitudes) resulting in 480 scenarios. Due to the limited number of identified faults in the outer-rise, the test set does not include this kind of earthquake. Their performance is rather verified against a single real tsunami event. Note that the training and test sets are based on the same simulation scheme for tsunami generation and propagation. Thus, additional tests for different modeling frameworks

are demonstrated through applications to three real events, which will be discussed later. The range of maximum amplitudes in the training and test sets, including those of real events, is shown in Fig. 3b, which denotes that all test set samples lie within the training set's scope.

**Prediction results on the test set**

The prediction is normally updated in real-time applications according to the data availability. Here we perform multiple predictions using maximum and mean tsunami amplitudes within three windows of 10, 15, and 20 min after the earthquake discarding the first 3 min of waveforms to circumvent seismic noise interference elaborated in the Method section. Examples of the type of inputs are illustrated in Supplementary Fig. 2, which also delineates the synthesized waveforms as observed by the bottom pressure explained in the Method section. To evaluate the results that predicted flow depths at inundated grids, we apply Eqs. (1)–(3) to all locations of flow depths ≥0.2 m and over all samples on the test set simultaneously.

As indicated in Fig. 4a, prediction errors in terms of geometric mean of amplitude ratios $K$ and its standard deviation $\kappa$ (often referred to as Aida's numbers, see Method) on the test set are $K = 1.08$ ($\kappa = 1.49$), 1.07 (1.50), and 1.02 (1.47) for the maximum amplitudes in prediction windows of 10, 15, and 20 min, respectively. Aida's numbers using mean amplitudes in 15 and 20 min are 1.10 (1.50) and 1.07 (1.47), respectively, comparable to the result using maximum tsunami amplitudes. However, the error is larger for the 10 min prediction window, reaching up to 1.22 (1.56). To assess the model performance more clearly, we also use an accuracy measure in percentage (Eq. (3) in the Method). The accuracies using maximum amplitudes as inputs are 92.6%, 93.5%, and 98.0% for prediction windows of 10, 15, and 20 min, respectively. For the same prediction windows, the accuracies of models with mean amplitudes are 82.0%,

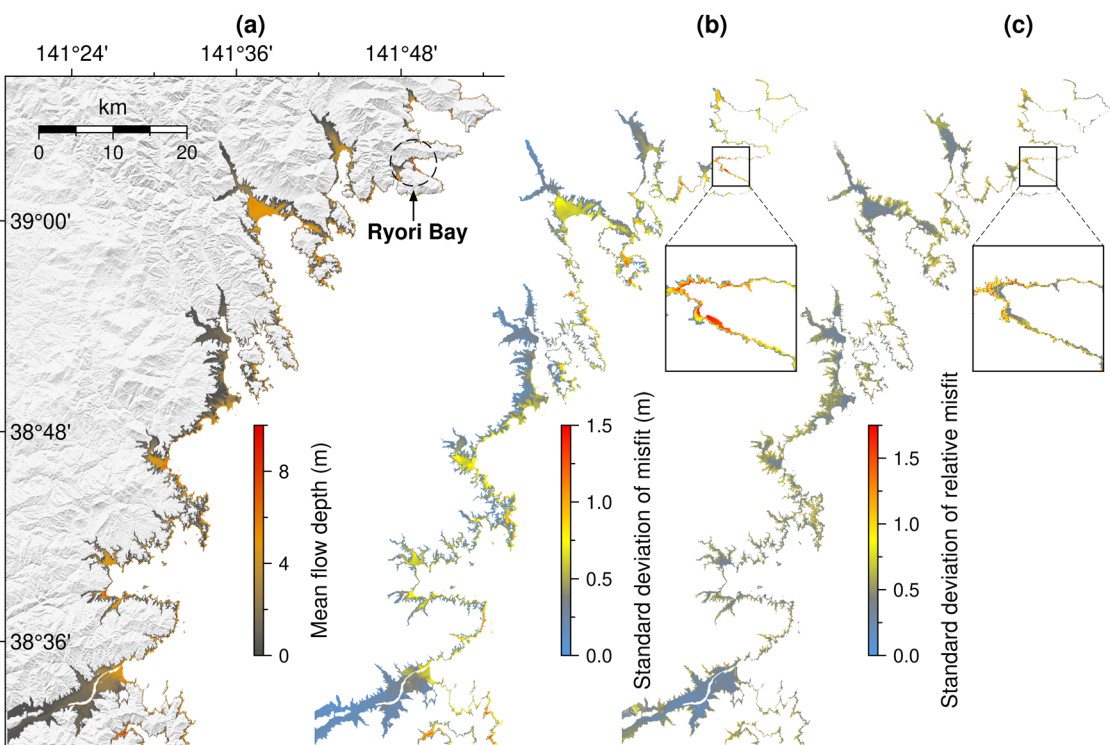

**Fig. 5 | The spatial variability of misfit on the test set. a** The mean flow depth. **b** The standard deviation of misfit. **c** The standard deviation of relative misfit. Calculations of **a**–**c** are based on all 480 samples on the test set.

90.9%, and 93.5%. These results indicate that both types of input exhibit comparable accuracy, but the input of maximum amplitude outperforms the mean amplitude particularly at a relatively short time window. For conciseness, hereafter, our discussion refers only to the results using maximum tsunami amplitudes as input within the 20-min prediction window.

Figure 4b shows the distribution of misfits (defined in the Method) of all samples on the test set. Like the other considered statistical measures, the evaluation is calculated for flow depths ≥0.2 m on the target. The figure indicates that the misfits are normally distributed with the median (50th percentile) centered around zero suggesting unbiased predictions. Figure 4c shows the goodness-of-fit statistic $G$ (see Method) for all samples at each specified earthquake magnitude. Generally, the $G$ values are larger, or the fits are worse for smaller magnitudes. This is an expected outcome because smaller earthquakes have a higher tsunami variability than the larger ones, as demonstrated in Fig. 3a. Similarly, our previous study[27] also found that tsunamis from smaller earthquakes are more difficult to estimate. Nonetheless, the present results are generally acceptable considering the misfit between the 15th and 85th percentiles is only <1 m (Fig. 4b). The overall median of $G$ ranges between 0.004 and 0.086 for all magnitudes. More details on the evaluation result using the goodness-of-fit statistic are tabulated in Supplementary Table 1.

We plot the comparisons between target and predicted inundation maps of randomly selected samples for megathrust earthquakes of magnitudes $M$w 8.2, 8.5, 8.8, and 9.1 on the test set in Supplementary Figs. 3a, b–6a, b. We also plot the residual or misfit between target and predicted inundations (Supplementary Figs. 3c–6c). Here, we perform the statistical evaluations over all inundated points for three flow depth ranges as described in the Method section. Overall, the predictions exhibit good agreement with the targets, as indicated by statistical evaluation results shown in the respective figure. Furthermore, we calculate the inundated area of both the target and prediction (shown in the figures). The predicted inundation areas also show good agreement with the targets, which is noteworthy considering the

predictions are derived from a regression of an exceptionally high dimensional output vector and not simply from the best scenario library selection as in previous studies[11,28]. These statistical evaluation results suggest that the model tends to better predict larger tsunamis, which is in line with the source variability factor discussed above and is relevant for early warning purposes.

**Spatial variability of misfits**

The spatial variability of misfits and relative misfits on the test set can be represented by its standard deviation over all samples (Fig. 5b, c). Ideally, the standard deviation value should be close to zero, which indicates the prediction skill consistency at the same location in various earthquake scenarios. However, this is not always the case because of the high degree of nonlinearity of inundation flow depths, particularly over complex regions. Higher misfits are apparent near shorelines with complex coastal geometry and steep cliff typically characterizing the southern Sanriku coasts[29] (Fig. 5b). While the large standard deviation of misfits occurs sporadically in relatively small areas, clustered large values of more than 1.5 m are located at Ryori Bay. Yamanaka et al.[30] suggested that the tsunami height inside the Ryori bay is most likely amplified by the bay resonance. In our case, the expected flow depths in this area are also high, as shown by the mean flow depth of the target of up to ~10 m (Fig. 5a), thus resulting in negligible relative misfits (Fig. 5c). This spatial variability of misfits can be thought of as one of the model limitations when approximating local effects from intricate coastal settings.

**Potential errors from malfunctioning stations**

In practice, bottom pressure sensors may fail to correctly record the data due to device malfunctions or other causes[31]. We anticipate this problem by arbitrarily selecting malfunctioning stations and setting the input value at these stations to zero. We gradually increase the number of malfunctioning stations from 10 to 140 with an interval of 10. Then, 100 random seeds are generated at each specified

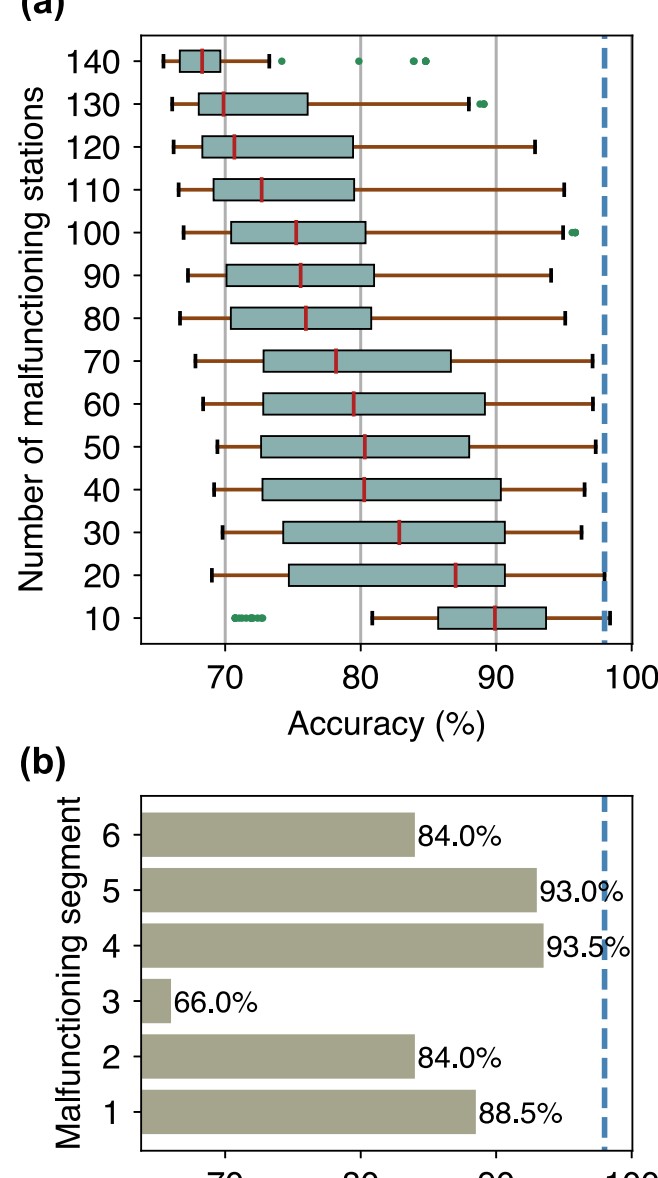

**Fig. 6 | Prediction accuracies relative to malfunctioning stations. a** Box plot of prediction accuracies with respect to the number of malfunctioning stations. **b** Prediction accuracies relative to a malfunctioning S-net segment. The dashed blue line marks the prediction accuracy without malfunctioning stations or segments.

S-net has six cabled segments, in which a segment consists of multiple observatories (see Fig. 1). Given such a configuration, it is possible that the entire stations on one or more segments fail simultaneously. However, here we consider only the malfunctioning of stations on a single segment (Fig. 6b). Similar to the previous experiment on the randomly selected malfunctioning stations, the accuracy measure is calculated over all samples on the test set. Figure 6b shows that accuracy of ≥84% can still be attainable when one of the S-net segments fails, except for segment three, which results in 66% of accuracy. This indicates that for our study area, segment three of S-net is crucial. The result also hints that many significant tsunamis from the stochastically generated earthquakes are possibly located within the coverage of segment three, which is consistent with the previous experiment result shown in Supplementary Fig. 7b.

**Application to real events**

To further assess the performance of the proposed method, we apply our machine learning-based model to real tsunami events generated by two megathrust earthquakes of the 2011 Tohoku-oki (Mw 9.0) and the 1896 Meiji Sanriku (Mw 8.1) events and one outer-rise earthquake of the 1933 Showa Sanriku (Mw 8.5) event. Among other tsunami episodes in Japan, destructions and characteristics linked with these three major incidents have been comprehensively reviewed[33,34]. More recent tsunami events recorded by S-net[2–4] were outside the training set distribution with no or minimum inundation traces, thus are not considered here. Since all three selected events occurred before the establishment of S-net in 2016, we synthesize the tsunami waveforms at S-net by simulating well-validated source estimates from previous studies[35–37]. Both tsunami sources of the 2011 Tohoku-oki[37] and the 1896 Meiji Sanriku[35] were inferred from tsunami data by an inversion analysis using Gaussian basis functions and finite faults, respectively. Whilst the source of the 1933 Showa Sanriku[36] tsunami was based on the best focal mechanism estimated from seismic data and validated using forward tsunami simulations. Supplementary Fig. 8 shows the tsunami sources from the studies mentioned above. Methods and setups used to produce all three source estimates of the real tsunami events are substantially different from those we apply to generate the data for training and test sets, particularly in terms of fault geometry and discretization. Therefore, these real tsunami events can provide deeper insights into the performance of our prediction model.

Simulation results of the real events using a physics-based model also serve as references for comparison with our proposed method. Figure 7 depicts tsunami inundation predictions for the 2011 Tohoku-oki event. Predictions of both inundation coverage and flow depth by the physics-based model and our method are shown in Fig. 7a, b (right panel), respectively. Despite the good agreement in the overall inundation coverage, ~2–3 m underestimations are visible around Rikuzentakata and Minamisanriku (Fig. 7c). Nevertheless, these occur at locations where flow depths from the physics-based model can reach up to 20 m. From the statistical evaluation results and calculated inundation area indicated in Fig. 7c, the performance of the machine learning-based model in predicting small flow depths (0.2 m ≤ h ≤ 1 m) is not as accurate as for the larger ones (h > 1 m), which is similar to the results on the test set. In addition, we also compare the predicted inundation heights from both models against real observations (Fig. 7a, b, left panel). To ensure consistency with the available observations, we convert our model output in flow depths into inundation heights by adding topography at observation points. For convenience, we only use Aida's number (Eqs. (1) and (2)) to evaluate the results. Our machine learning-based model results in $K(\kappa)$ of 0.99 (1.39), which is comparable to the physics-based model of 0.96 (1.40). This further demonstrates the model's capability to accurately predict significant or larger flow depths or inundation heights as predominantly found in real observations.

number of stations to represent various combinations of stations. For convenience, we only use the accuracy metric (Eq. (3)) based on the 20-min prediction window (Fig. 6a). In general, 10 malfunctioning stations slightly reduce the accuracy compared to that of the result when all S-net stations are available at 98%. However, an extreme outlier with an accuracy of 70.7% is apparent, suggesting that the model is susceptible to inconsistent inputs relative to the training set distribution commonly attributed to the neural network predictions[32]. On the contrary, prediction accuracy of 84.8% can still possibly be achieved with 140 malfunctioning stations. We plot the corresponding station locations in Supplementary Fig. 7. Although there is a better way to appraise the role of observation points in the prediction[27], this experiment may imply prominent S-net stations needed to be maintained to sustain accurate predictions.

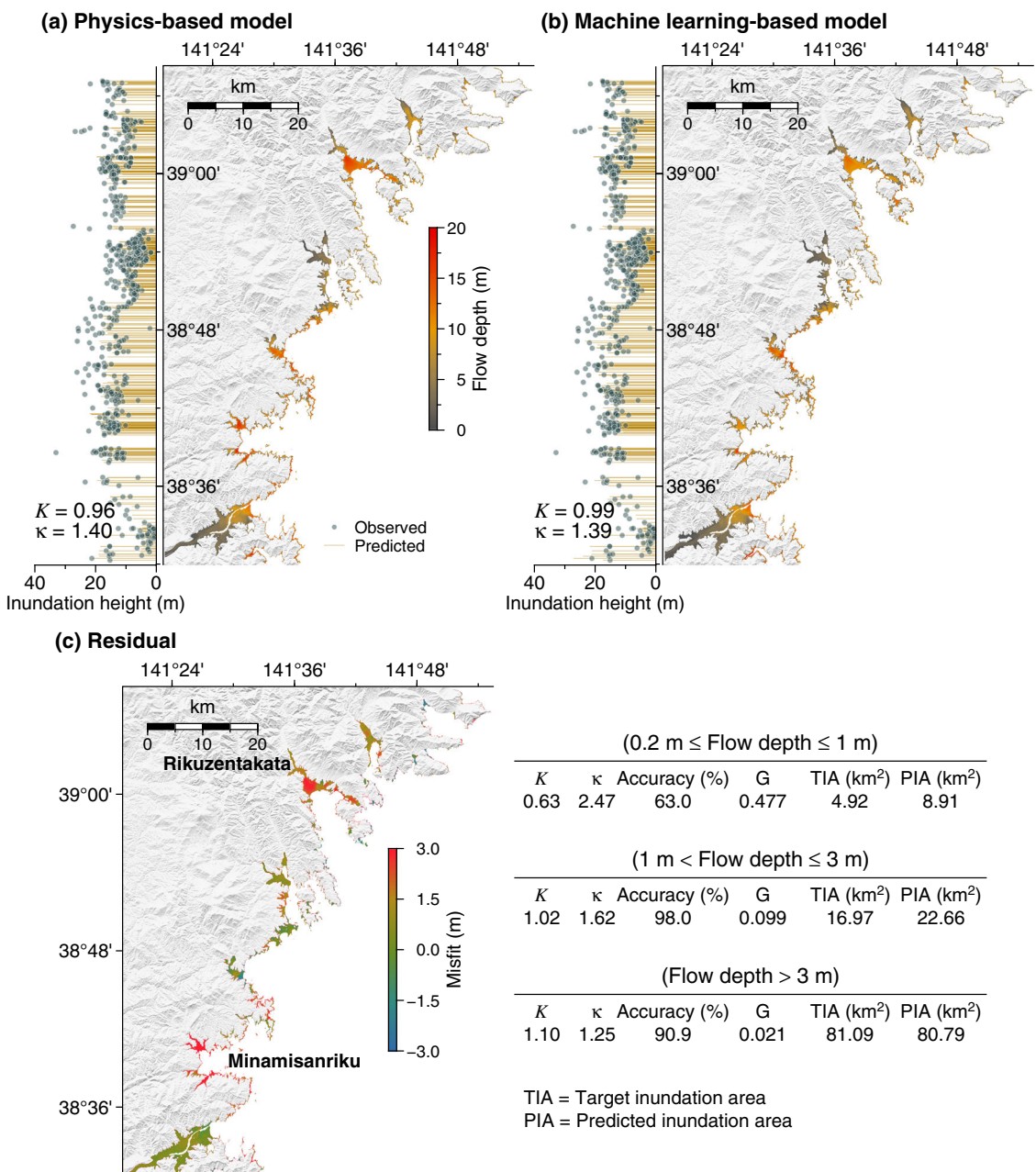

**Fig. 7 | Prediction result and evaluation for the 2011 Tohoku-oki event.**
**a** Physics-based model. **b** Machine learning-based model. Inundation map of flow depths at locations of interest (right panel) and comparisons between observed and predicted inundation heights (left panel). **c** The misfit and statistical evaluations of **b** relative to **a**.

An analogous trend of performance improvement for predictions with larger flow depths is also attained for the 1896 Meiji Sanriku event (Supplementary Fig. 9). However, the overall accuracy for the respective specified level of flow depth (Supplementary Fig. 9c) is inferior compared to the prediction for the 2011 Tohoku-oki event. This is partly because the source model of the 2011 Tohoku-oki event[37] (Supplementary Fig. 8a) was based on inverted surface displacements rather than fault slips. Therefore, the initial condition of the tsunami well matches the surface displacement profile from the more modern and complex plate geometry estimate[20] used to generate the training set (see Fig. 2 for examples). In contrast, the source model of the 1896 Meiji Sanriku event[35] (Supplementary Fig. 8b) was calculated from slips over a simplified fault geometry. Furthermore, the quantity and quality of data used to infer the 2011 Tohoku-oki source were more reliable than the 1896 Meiji Sanriku event. Notwithstanding the above issue,

predictions for larger flow depths are still within or close to the expected range of criteria for satisfactory model performance[38]. The prediction accuracy in terms of $K(\kappa)$ of our model relative to the real observed inundation heights is 0.96 (1.41), while the physics-based model is 1.03 (1.39).

The 1933 Showa Sanriku event results are shown in Supplementary Fig. 10. The overall performance of the machine learning-based model is quite similar to the 1896 Meiji Sanriku event (Supplementary Fig. 10c). However, here the main issue is likely caused by the limited number of outer-rise event scenarios in the training set. Moreover, the tsunami source for the outer-rise earthquake of the 1933 Showa Sanriku event was estimated using a uniform slip, a simpler source model than the other two events. Although the model can be sufficient for the far-field[39], its limited accuracy in the near-field adversely affects our ability to obtain a reference model that can well reproduce the

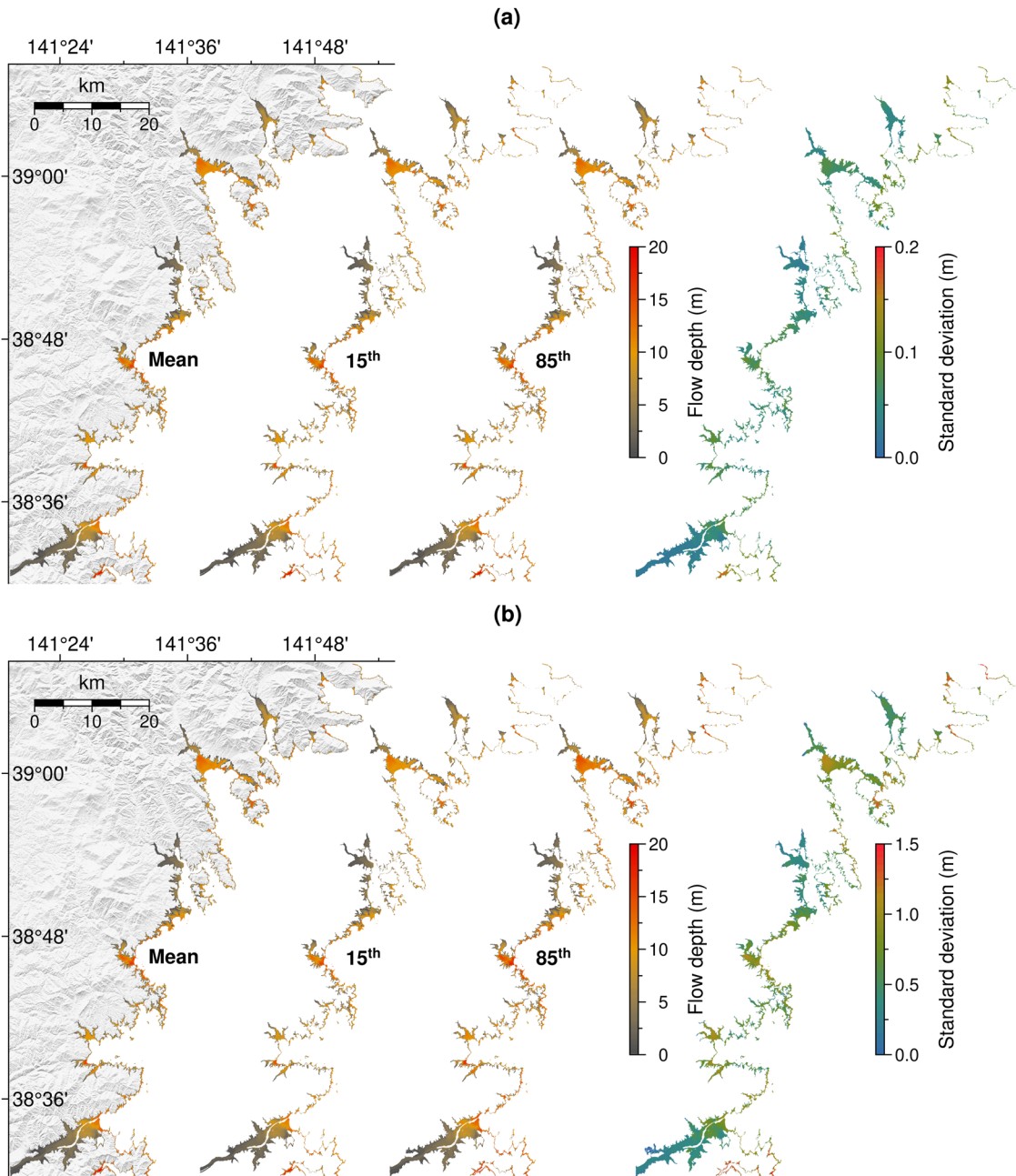

**Fig. 8 | Prediction uncertainty associated with observational errors.** The mean, 15th percentile, 85th percentile, and standard deviation of 100 ensemble flow depth predictions from the perturbed input values for the 2011 Tohoku-oki event. **a** Using the 0.01 m standard deviation of input perturbations. **b** Using the 0.1 m standard deviation of input perturbations.

observed tsunami heights. Comparisons with real observed inundation heights result in the $K(\kappa)$ values of 0.92 (1.81) and 0.95 (1.59) for our method and the physics-based model, respectively. Ideally, more outer-rise faults should be incorporated into the data, albeit challenging as their identifications rely on the limited historical records or marine seismic observations and surveys[22–25,40].

**Prediction uncertainty due to observation errors**

Although the proposed method can be free from the uncertainty of source estimates, the predictions are still subjected to other sources of uncertainty. The previous result on the spatial variability of misfits, to a lesser extent, reveals the prediction uncertainty related to the model and local effects. Here, we introduce another potential source of prediction uncertainty corresponding to input or observation errors.

However, since historical records of tsunamis at S-net are limited to properly quantify the observation error, we assume random perturbations of input sampled from a Gaussian distribution with zero mean and standard deviations of 0.01 and 0.1 m (Supplementary Fig. 11). Such levels of error are reasonable considering that S-net was able to detect small tsunamis of less than one centimeter[3,4]. As an example, we run 100 ensemble predictions for the 2011 Tohoku-oki event incorporating the above perturbations to the input of maximum tsunami amplitudes. The prediction mean, percentiles, and standard deviation from the 100 ensembles are shown in Fig. 8. Figure 8a shows that the perturbed inputs with the error standard deviation of 0.01 m lead to the variability of predicted flow depths with a maximum standard deviation of 0.2 m. In comparison, the maximum standard deviation of predicted flow depths can reach up to 1.5 m when the inputs are

perturbed with the 0.1 m of error (Fig. 8b). In addition to demonstrating the prediction uncertainty and sensitivity, the results also highlight the nonlinear relationships between inputs and predictions.

### Computation time

We invest a considerable computational effort in preparing the data through numerous simulations normally required in precalculated-based tsunami predictions[11,13,14,28]. The neural network training is another computation carried out in advance of the real-time application. The training time of our model is ~65 min using a single graphics processing unit (GPU) of the NVIDIA GeForce RTX 2070. However, we utilize a traditional central processing unit (CPU) for a fair comparison in real-time computation because our physics-based model is programmed only for CPUs. Without a parallelization, the computing time (on Intel i7-10875H @ 2.3 GHz and 16 GB of memory) to deliver the inundation prediction shown in Fig. 7a using the physics-based model is approximately 30 min. With the same machine, the inundation prediction by our method (Fig. 7b) is obtained within a mere 0.05 s, which is ~99% more computationally efficient than the conventional approach. The remarkable computing speed will also facilitate further improvements in the prediction when a higher bathymetry and topography resolution is considered.

Besides the real-time computation, the time required to retrieve the data equivalent to the prediction window should also be factored in, corresponding to the length of data used to invert the source estimate in the physics-based model. The source estimate considered in this study to initiate the inundation prediction (Fig. 7a) was partially inverted from tsunami data of more than 1 h after the earthquake since it was not meant for a real-time prediction[37]. Due to the remarkable S-net coverage, real-time source estimates can be obtained within ~8 min[41], which was difficult to achieve in the absence of S-net[42,43]. However, improving the result through sequential predictions with a continuous data stream can be troublesome, as the physics-based model needs to be run for each updated source. In contrast, our prediction result can rapidly be updated when more data are available. The total time to make the predicted inundation shown in Fig. 7b is ~20 min, similar to the prediction window, as the real-time computation is negligible.

## Discussion

The offshore celerity of near-field tsunamis likely outpaces our ability to produce an accurate prediction promptly when employing traditional methods[44]. This study demonstrates that a machine learning-based model can be used as a potential surrogate for the conventional physics-based model to predict near-field tsunami inundations in real time. Moreover, we configure our model to take into account considerably wider inundation zones than that commonly adopted in tsunami modeling[11,13,14,28]. The model can also be feasibly extended to encompass more areas without being compensated by a significant increase in the real-time computational cost. While the computational speed of the algorithm and the prediction accuracy are promising, we recognize there are several outstanding issues needed to be resolved prior to the actual practice in the operational forecasting system.

A real-time operational tsunami inundation prediction algorithm has been integrated into the US National Oceanic and Atmospheric Administration's tsunami forecast and warning system[45,46]. The system aims and has been successfully implemented to predict far-field tsunami events. This study offers an alternative to the existing methods for more challenging near-field tsunamis, by which a timely prediction should be made under a restrictive time constraint. Therefore, in conjunction with the accuracy, the prediction rapidness is regarded as the main feature of our proposed method. Compared to the conventional tsunami modeling approach, our method has effectively taken advantage of S-net's extensive offshore observation network to address the aforementioned issue. However, the lack of real significant

or large tsunami records presently registered at S-net renders further evaluations against real tsunami data a seemingly impossible task at the moment.

Another limitation of our method is related to spurious predictions when many S-net stations malfunction leading to inconsistent inputs with the training set distribution, as demonstrated by our synthetic experiments (Fig. 6). Furthermore, although we have considered a wide range of tsunami scenarios, we cannot rule out the possibility of unprecedented events that are outside the expected range. Consequently, expert judgments for data quality assurance are needed to prevent unexpected outcomes. An automated data quality control can also be implemented, which is similar to that used in the seismic[1] and atmospheric[47] observations. On the other hand, the experiment also hints that a reasonable level of accuracy can be attained using a smaller number of stations. A method to search for optimized subsets of stations[27,48] can be combined with the neural networks model. Then, multiple prediction models are developed for separate optimal subsets, facilitating further prediction uncertainty quantification. This also allows for the realization of the method in regions with a limited number of offshore tsunami observations.

Recently, cost-effective designs of regional tsunami observation networks have been proposed in many tsunami-prone countries, e.g., south of Java of Indonesia[49], north of Chile[50,51], and the Mediterranean Sea[52,53]. Additional inputs from seismic or geodetic data may also be harnessed to complement the scarce offshore tsunami observations[11,14,15]. Moreover, the versatility of neural networks enables a wide variety of ways to adjust the input structure straightforwardly; as such, it can be built to make use of time series at each station[15,16] instead of a single representative value as in our case. This, too, would be particularly beneficial for a prediction model with minimum observation points. Another type of input that can be extracted from tsunami data is the arrival time. Although it is not considered in this study, such information will be useful in providing a fast first warning product. The contribution of tsunami arrival times in reducing the prediction uncertainty has been reported in a previous study[54].

One of the essential aspects of tsunami forecasting is related to prediction uncertainty. Quantifying the prediction uncertainty is a challenging task, especially in real time. Machine learning-based methods can facilitate such a task owing to their computational speed. In our case, the 100 ensemble predictions can be obtained within seconds using a single GPU with the above specifications. However, we note that the uncertainty reported in this paper may not reflect the total prediction uncertainty. For example, the model uncertainty of machine learning-based approaches can be estimated through prediction ranges from different initial weights[16], which is not considered here. A more comprehensive prediction uncertainty of tsunamis is typically incorporated in a probabilistic tsunami hazard assessment (PTHA)[55,56], in which both epistemic and aleatory uncertainties are accounted for. More recently, Selva et al.[57] successfully combined real-time observations with long-term hazard estimations from the PTHA to rapidly quantify the uncertainty in real-time tsunami forecasting. Future developments of machine learning-based tsunami predictions should be directed toward a more explicit quantification of total prediction uncertainty.

Lastly, advances in machine learning have led to various types of neural networks with their respective architectural building block[58]. However, a comparison study of different neural network models applied to our case is of our future interest and is beyond the scope of this paper. In addition, a better hyperparameter tuning should be carried out, possibly through an optimization[59], which will also be considered one of the study's future directions. Herein, we emphasize the novel way to leverage the present world's most extensive offshore tsunami observing system readily available in the study area. In addition to the more widespread adoption of offshore tsunami observing systems, we foresee that underpinned by the accelerating progress in

computer science, substantial improvements of the proposed method are most likely expected in the near future, thus enhancing the tsunami mitigation capability globally.

## Methods

### Stochastic slip model

Guided by the SLAB 2 model[20], we discretize the plate interface of the Japan Trench subduction zone with the downdip limit of 40 km into 240 curvilinear patches or subfaults with approximate sizes vary from ~20 to ~40 km$^2$ as shown in Fig. 2. Then, we generate a large number of hypothetical megathrust earthquakes with magnitudes ranging from $M$w 8.0–9.1 (interval of 0.1) using a method to characterize the complexity of earthquake slip represented by a spatial random field based on a von Karman autocorrelation function[19]. Initially, the random slip is calculated on a rectangular fault of 1 km × 1 km grid size with the length and width estimated using a magnitude-area scaling relation[60]. Subsequently, we interpolate and project the resulting slip into the curvilinear patches with arbitrary rupture area while preserving the moment magnitude. To calculate the coseismic displacement from the slip assumed to be similar to the sea surface deformation, we use an analytical solution for triangular dislocations in a half-space[21]. Therefore, each patch consists of a pair of triangular meshes with the same slip amount[27].

### Physics-based tsunami simulation

The propagation of tsunami waves can be governed by the depth-averaged shallow water equations comprising the linear and nonlinear components reflecting offshore and nearshore behaviors, respectively[5]. Normally, the equations are solved numerically using a nested grid system with increasing domain and grid sizes toward the sea for computational efficiency. A moving coastal boundary condition is used on the smallest domain to accommodate the inundation processes, while fully reflective coastal boundaries are enforced on the remaining domains. This study uses a well-validated numerical tsunami simulation code called JAGURS[6,7,61]. We divide our model domain into a four-level nested grid system with sizes of 30, 10, 3.33, and 1.11 arcsec (Supplementary Fig. 12a). We run the simulation for 3 h with a time step of 0.5 s and bottom friction achieved using a Manning coefficient of 0.025 s/m$^{1/3}$. A crustal deformation effect manifested by landward subsidence may elevate the inundation[62]; on the contrary, landward uplift may reduce the inundation potential; these effects are accounted for in the model. Bathymetry and topography are resampled from the General Bathymetric Chart of the Oceans (GEBCO_2020 Grid) with 15 arcsec grid resolution[63], the Japan Hydrographic Association's M7005 bathymetric contours, and the digital surface model of the Japan Aerospace Exploration Agency with 30 m grid spacing[64]. Supplementary Fig. 12b shows the resampled bathymetry and topography on the highest resolution domains. This physics-based model is used to simulate tsunami Green's functions, hypothetical tsunami scenarios, and real tsunami events discussed in this study.

### Tsunami Green's functions

Tsunami Green's functions are a collection of synthetic tsunami waveforms at observation points originating from multiple subfaults with slip typically set to 1 m. The method builds on the premise that the tsunami waveforms associated with the actual earthquake slip can be formed from a linear superposition of these synthetics[26]. The assumption is valid in offshore regions where the linearity of tsunami waves prevails. Generally, this technique is used to invert the slip of a tsunamigenic earthquake from observed tsunami waveforms[26,35]. Here, however, since the slip is the given parameter, we utilize Green's functions summation technique[27] to synthesize tsunami waveforms at S-net stations. Such an approach is computationally more efficient than the standard forward model simulation when the number of subfaults is less than the number of considered source scenarios.

Referring to the fault model in the preceding stage, we construct the tsunami Green's functions from the 240 subfaults. We use them to generate tsunami waveforms from 500 earthquake scenarios at each magnitude interval for the input variability analysis (see Fig. 3a).

### Input types and variability

To analyze the variability of inputs expected to give clues on the sufficient number of source scenarios in our application, we apply Green's functions to 500 stochastically generated slip models for each of all considered earthquake magnitudes. This results in a series of synthetic water surface fluctuations at S-net stations as observed by the bottom pressure sensor. We subtract the corresponding initial displacement at station locations from the resulting tsunami waveforms to mimic pressure waveforms. Accordingly, the process inflicts a constant offset in the pressure readings signifying the coseismic permanent seafloor displacement effect naturally occurred at stations inside the source region (Supplementary Fig. 2). Such an effect is often regarded as one of the main complications in utilizing bottom pressure data for tsunami studies[65,66], but not necessarily for our case. Conversely, the short-period background noise has been reported to be very small as S-net has successfully detected millimeter-scale tsunamis[3,4]. However, bottom pressure data can be interfered with by a low-frequency component of seismic waves that are difficult to distinguish from tsunami signals, particularly at stations near the earthquake focal area[67]. Similar to a previous study[68], to avoid using corrupted data, we omit the first 3 min of pressure waveforms, during which the seismic wave disturbance predominantly occurs.

Once all bottom pressure waveforms at S-net have been generated, we pick the maximum amplitude located between 3 and 20 min after the earthquake for the subsequent analysis (Supplementary Fig. 2). We then calculate the coefficient of variation, defined as a standard deviation divided by a mean of samples. These samples correspond to the 500 slip models with the associated maximum tsunami amplitudes at 150 S-net stations. At each specified earthquake magnitude, the coefficient of variation is calculated at every increment of samples from 2 to 500. A previous study has conducted the same analysis but for a different property of tsunami[27].

### Neural networks configuration

We apply simple fully connected neural networks acting as a non-parametric regression model[69] as illustrated in Fig. 1. The input layer consists of 150 neurons corresponding to the number of S-net stations. Each input layer neuron constitutes a single value of observed tsunami properties (e.g., maximum or mean tsunami amplitude) at the respective station, and the output neuron represents flow depths on inundated grids. As the number of inundated grids varies with scenarios, we fix the grid points equivalent to the number of neurons in the output layer according to the maximum inundation extent available on the training set. In such a way, flow depths on inundated grid points by larger tsunamis are set to zero for relatively smaller tsunami scenarios. The final number of grids is remarkably large, comprising of 214,356 grid points that are then flattened into a single column vector of the output layer. The grid coordinates are provided in Supplementary Dataset 1. After some trial and error, we find that two hidden layers, each with 150 neurons, suffice to meet the expected result for our application.

A rectified linear unit (ReLU) activation function is used in the hidden layers with a 20% dropout rate for better generalization[70,71]. As the flow depths value is always positive, we also find that implementing the ReLU activation function in the output layer helps suppress erroneous inundation patterns. For our neural networks model training, we use the Adam optimization algorithm[72] with the He normal initialization[73] and a batch size of 20. A mean squared error commonly applied to regression tasks is used as the loss function. However, to assess the model performance more comprehensively, we use several

statistical measures that are widely adopted and relevant for tsunami cases described in the subsequent section. We opt for the final model that gives the smallest error on both training and test sets after 2000 epochs. The neural networks model is implemented using the TensorFlow library[74], with most of the model settings and parameters set to default.

## Model evaluations

We denote $O_i$ as observed or target and $S_i$ as predicted or simulated flow depths at $i$th location. The simplest measure of accuracy is defined in terms of a misfit, $O_i - S_i$, and a relative misfit, $(O_i - S_i)/O_i$. We also consider Aida's number[75] commonly used to evaluate the performance of simulated tsunamis relative to observations[11,28,76]. With this statistical measure, the prediction accuracy over all considered locations ($N$) is appraised according to a geometric mean ratio ($K$) between the observed and simulated values, together with its standard deviation ($\kappa$) formulated as follows:

$$\log K = \frac{1}{N}\sum_{i=1}^{N}\log(O_i/S_i),\qquad(1)$$

$$\log \kappa = \left[\frac{1}{N}\sum_{i=1}^{N}(\log(O_i/S_i))^2 - (\log K)^2\right]^{1/2}\qquad(2)$$

The prediction results are generally acceptable when the values of $K$ and $\kappa$ are within or close to the suggested criteria for satisfactory model performance, which are $0.8 < K < 1.2$ and $\kappa < 1.4$[38]. An underestimation and overestimation of the observation are indicated by the $K$ value larger and smaller than 1, respectively. In addition, following previous studies[76,77], an accuracy measure in percentage can be formulated from the $K$ value as follows:

$$\text{Accuracy (\%)} = \begin{cases} 1/K \times 100\%, K \geq 1 \\ K \times 100\%, K < 1 \end{cases}\qquad(3)$$

To further assess the performance of our proposed models, we also implement a statistic to define the goodness-of-fit ($G$) between observed and predicted values modified from a previous study[78],

$$G = 1 - 2\frac{\sum_{i=1}^{N} w_i^2 O_i S_i}{\sum_{i=1}^{N}(w_i O_i)^2 + \sum_{i=1}^{N}(w_i S_i)^2},\qquad(4)$$

where $w_i$ is a weight proportional to the absolute value of $O_i$ at evaluation point $i$. We limit the minimum weight to be one-third of the maximum value of $O_i$ over all $i$, so that the analysis focuses on higher flow depths. Since flow depths are always positive, the $G$ value ranges between 0 and 1, in which lower values indicate a better fit.

For the test set, we perform all the above statistical evaluations only at locations with observed flow depths $\geq 0.2$ m. For the real events and some selected samples, we further categorized the evaluations based on criteria for tsunami advisory or warning issuance by the Japan Meteorological Agency (https://www.data.jma.go.jp/svd/eqev/data/en/tsunami/tsunami_warning.html). The evaluations are conducted separately for flow depths ($h$): $0.2$ m $\leq h \leq 1$ m, $1$ m $< h \leq 3$ m, and $h > 3$ m. The purpose of this analysis is to assess the model performance in relation to a practical tsunami early warning system, such that the prediction accuracy is focused on regions with significant tsunamis or high flow depths.

## Data availability

The bathymetry data were obtained from https://www.gebco.net/ and https://www.jha.or.jp/en/jha/. The topography data were downloaded from https://www.eorc.jaxa.jp/ALOS/en/aw3d30/index.htm. The plate interface depth is available at https://github.com/usgs/slab2. The observed tsunami inundation heights are from the tsunami trace database provided by the International Research Institute of Disaster Science, Tohoku University, Japan, available at https://tsunami-db.irides.tohoku.ac.jp/.

## Code availability

The tsunami simulation code of JAGURS is available at https://github.com/jagurs-admin/jagurs. The method used to generate stochastic slip realizations can be downloaded from http://equake-rc.info/cers-software/. The code to calculate the triangular dislocations in a half-space is provided as supporting information to the paper by Nikkhoo and Walter[21]. The neural networks model was based on the TensorFlow library downloaded from https://anaconda.org/conda-forge/tensorflow. All figures were made using the GMT software[79] (https://www.generic-mapping-tools.org/download/) and the Matplotlib library implemented in Python (https://matplotlib.org/).

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

## Acknowledgements

The work is part of the RIKEN Pioneering Project "Prediction for Science". I.E.M. is funded by the Japan Society for the Promotion of Science (KAKENHI No. 22K14459). We would like to thank Tatsuhiko Saito for providing the source estimate of the 2011 Tohoku-oki tsunami. We appreciate the comments and suggestions from reviewers that substantially improved the quality of our manuscript.

## Author contributions

I.E.M. conceived the study and performed the computations. N.U. and T.M. led and supervised the project including results interpretation. A.R.G. undertook the data collection and was involved in the numerical simulations. K.S. contributed to the result interpretation and analysis. I.E.M. wrote the manuscript together with all authors.

## Competing interests

The authors declare no competing interests.
