## [Peer Review File · Nature Communications]

Machine learning-based tsunami inundation prediction derived from offshore observationsREVIEWER COMMENTS

Reviewer #1 (Remarks to the Author):

Summary:

This paper proposed a machine learning method to forecast tsunami inundation (height and coverage) based on real-time information from offshore tsunami observing stations. The proposed method has the main advantage of estimating tsunami inundation without knowing the earthquake source model or initial tsunami source. Recently, machine learning techniques have been widely used in various research fields appealing to the remarkable ability to solve complex problems. The application of machine learning is not zero, but it is not so progressed in the tsunami research fields. In addition, predicting tsunamis in real-time is a great issue in tsunami disaster mitigation. Accordingly, the theme of the paper is very interesting. The paper is also well-written and structured; therefore, I recommend this paper for publication in Nature Communications after the authors addressed the following comments and questions:

Comments and questions:

1. P9, L175. The authors mentioned that the number of testing scenarios is 420. If 20 hypothetical megathrust earthquakes are generated from each magnitude (12 magnitudes), the total number of testing scenarios is 240?
2. Why do the authors only consider 33 fault models for the outer-rise earthquakes? Why don't the outer-rise scenarios be defined in the same manner as megathrust scenarios? See <https://doi.org/10.1002/2014JB010958> and <https://doi.org/10.1007/s00024-019-02364-4>
3. The input of the machine learning model for the prediction is the maximum and mean tsunami amplitude, and eventually, the maximum value provides better results (for short time windows). Why do the authors use those values, not a whole part of waveforms? In this manner, I expect the prediction results to be much better.
4. P3, line 65, typo "uncerteinties"?

Reviewer #2 (Remarks to the Author):

The manuscript "Machine learning-based tsunami inundation prediction derived from offshore observations" describes a AI-based approach to emulate tsunami simulations of inundation over large target areas based on offshore observations. I think that the presented research presents interesting results. However, I have some concerns that I describe in the followings, and that I think the authors can fully address in a new version of the manuscript.

My main concern is about the selected procedure to demonstrate the goodness of the results. The adopted procedure is not very convincing to me, and I think that more must be done. In particular, the authors use, as main metric to evaluate results (reported also in the abstract, line 27), the "root mean square error" for several tsunami intensity measures (introduced in line 187). To my understanding, this is the root mean square of the residuals between the physics-based and the machine-learning-based tsunami simulations, averaged over all target points and all considered scenarios (please, see minor points regarding its definition). My fear is that this value may be strongly biased toward small values by a large number of target points (e.g. all the ones far from the coast) for which predicted and reference values are both practically equal to 0 (no tsunami). All target points far from the coast are in this situation, at least for all smaller magnitudes. This may introduce a large number of zeros into the evaluation of the root mean square that decreases a lot its final value. In practice, if this is the case, the root mean square error can be made as small as we wish simply by adding more points inland, where no tsunami will occur. This is somehow explicit in figure 6, where all points far from the coasts show values close to 0, while all points close to the coast have larger values. This means that the reported good

numbers may be dominated by forecasts in areas in which the tsunami does not occur at all (or it is negligible), and small average residuals may be obtained almost independently from how good the model works where the tsunami actually occurs. Thus, I think that it could be useful:

- To focus the comparison on the areas in which the tsunami is expected, considering much smaller areas for smaller magnitudes (defined, for example, using as reference the physics-based simulations, and considering only points where the tsunami exceeds a lower threshold for inundation depth, e.g. 0.1 m).
- To propose a comparison also using relative differences, instead of absolute differences. Indeed, 1 m of misfit is maybe acceptable if 10 m are expected, while it may be not acceptable if 0.5 m is expected.
- To report also the average of residuals (and not only the root mean square), to evaluate also the existence of potential biases in the predictions. To this end, also the statistics of the Aida's number (comparing physics-based and machine-learning-based results) could be useful.
- Given the importance of inundation areas, I suggest to introduce also some quantitative test about the correspondence of modelled inundation areas. In line 212, a "nearly perfect agreement" is stated for inundation areas, but no quantitative discussions about this quantity has been presented.

To improve testing, it may be also useful to introduce some quantitative statistical test to rule out the existence of significant biases, for example demonstrating that the average misfit in the areas with significant tsunami is compatible with 0 (see, for example, the tests proposed in Davies 2019 - *Geophys. J. Int* - or in Selva et al. 2021 - *Nat. Comm.*), potentially keeping separated the scenarios for the different magnitudes.

A second important concern is about the reported discussion about the "prediction uncertainty". To my understanding, what the authors call "uncertainty" or "prediction uncertainty" (see, for example, line 344 or Fig. 6) is the average difference between physics-based simulations and machine-learning-based simulations. In my opinion, this is not the uncertainty on predictions, but the misfit between the two models. Prediction uncertainty is instead about the difference between models and the actual tsunami, and should include both input and modelling uncertainty for each individual tsunami. Thus, the discussed uncertainty is only a part of the total prediction uncertainty.

More specifically, most of the tests are made comparing physics-based and machine-learning tsunami models, and both training and testing datasets are produced using exactly the same tsunami model (same source, from SLAB2; same tsunami simulation scheme, etc., please see also minor comments about this). The limited comparison with real data demonstrate that physics-based and machine-learning methods perform similarly in predicting real data (lines 260-263, 285-289), but none of them is a "perfect prediction" (whatever this may mean). To well reproduce physics-based models is indeed an important results, but it is not an evaluation of the prediction uncertainty.

Therefore, on the one side, I suggest to use expressions like "modelling misfit" (or something similar) instead of "uncertainty" or "prediction uncertainty" throughout the manuscript. On the other side, I suggest to actually discuss the problem of prediction uncertainty, at least in the Discussion, as this is an important issue for the applicability and thus the potential impact of the proposed method. Machine-learning-based simulations are indeed very quick, and this may give the opportunity in the future to actually quantify prediction uncertainty using ensemble of different simulations (like proposed, for example, in Selva et al. 2021 - *Nat. Comm.* - or Giles et al, 2021 - *Front. Earth Sci*), quantifying the variability of the deterministic predictions over similar initial conditions or alternative modelling frameworks. This is somehow suggested in lines 367-369, but it is not properly discussed in the present version of the manuscript.

A third important concern is about the differences between the presented procedure and the one recently published in Makinoshima et al. (2021 - *Nat. Comm.*), which share method (CNN), input data, and target area. I think that probably the authors have good points on their side, however I think that a deeper discussion is needed, to better demonstrate and highlight the effective novelties of this manuscript. For example, Makinoshima et al. state that the their CNN simulations takes 0.004 s in a single CPU for a single target point, meaning that their method can be adopted

in parallel for 100 or more points, still obtaining results in a fraction of second in a single CPU. In addition, differences between the adopted Neural Network configurations, or eventual differences in the pre-processing of the observation data (which are the same here and in Makinoshima et al.), should be further commented, in my opinion.

Finally, more attention should be devoted to discuss the potential practical applicability of the procedure in a real system, enriching the discussion in this sense. The arguments reported in the introduction (lines 97-101) are poorly discussed and not supported by specific literature, for example about how to inform in such a short time, or how to use such information during an ongoing evacuation. In addition, a link to the uncertainty on the forecast should be probably added, as already commented above.

Other minor points are:

- Lines 103-104: please report the number (or the order of magnitude) of the target points. It would be interesting, for reproducibility of the results, to have the target grid as supplementary material.
- Line 173: 33 is a very limited number to describe the potential variability of such sources. I suggest to comment this. I also suggest to report the list of the actual source parameters as supplementary material.
- Line 175: the 480 scenarios used for testing are produced using the same simulation scheme used for training the model, both in terms of source and simulation scheme. This has to be better stressed, as it has consequences on how to interpret the results.
- Line 187: please define explicitly the "root mean square error", for example with an equation in Method, section "Model Evaluation"; I struggled a bit to understand its actual definition.
- Lines 211-212: I suggest avoiding words like "small" or "big". Also, please the same names in all the manuscript. Here, with "errors" you probably mean "root mean square error", and it may be useful to introduce a specific symbol for it, as it is widely used in the manuscript.
- Line 212: "nearly perfect" is too generic. Maybe inundation area should be properly tested, as suggested above.
- Lines 235-236: consider to explain here why none of the events recorded by the network (which are mentioned before) cannot be used here.
- Lines 242-249: are such sources modeled in the same 3D fault surface used before, or this is completely independent?
- Line 247: please, be more quantitative and report more details about such substantial differences
- Lines 261-262 it may be useful to specify, here or in methods, that K close to 1 means no biases in the estimation, and that numbers $<$ or $>$ 1 correspond to over and under-estimations
- Fig.5: please, report also the difference between the physics-based and machine-learning-based results. In this way, it is almost impossible to evaluate. The comparison with data is not very effective with this representation. Maybe it can be enlarged and put in a separate panel, also reporting some histograms, etc
- Line 290: I suggest to change name to this section. I suggest something related to the spatial variability of misfits
- Line 291-292: with respect to what earlier is called "root mean square error", the only difference here, is that you do not average in space, correct? If so, please specify
- Line 302: the mean is probably biased by all points with 0 misfit. Maybe, it is more appropriate to report some percentile (e.g., 90th percentile).
- Line 325: "decent" is not very quantitative
- Lines 356-358: see comment to lines 235-236
- In figures 3-6 of the supplementary material, please add a plot of the misfit, in order to better appreciated differences
- Lines 494-505: please, check this part, as it seems to me that there is an error here, between predicted P and observed (O) values. Note that O is not present in the equation.

Reviewer #3 (Remarks to the Author):

General Remarks

The manuscript of Mulia et al. describes the design and preliminary test of a tsunami forecasting

system based on a neural network approach. The tensor flow based neural network is trained with approx. 3100 synthetic tsunami scenarios and is then tested by a number of simulated events - partly informed by field data - not contained in the training set. The results are compared with simulation results obtained by a physics-based forward simulation and the authors find that their neural network is 99% more computationally efficient, while producing acceptable deviations in the inundation flow depth. In this respect the findings are relevant and timely, since a number of articles in the past few years developed similar important progress in near field tsunami forecasting.

I have a few fundamental questions that I'd like to see addressed before I feel the manuscript acceptable for publication in Nature:

1. The results or quality metrics of the forecasts are basically all synthetic. While the ground truth is based on hindcasts of real events, the input data for the neural network are all synthetic. Additionally, the error between the physical and the learned results are also reported in terms of comparing synthetic values - in weather forecasting this would be called a re-analysis. While this is an important first step towards a real forecasting system, it is nothing that marks a break-through and in particular is still prone to large uncertainty.

2. As a follow-up to my first point, I think the sensitivity analysis of the newly designed forecasting approach does not cover relevant possibilities. While the authors do consider the failure of individual inputs (basically, failing random individual sensors), more relevant realistic scenarios are not considered. Due to the design of the S-net sensor network in serial lines of sensors, in case of an earthquake rupture it is quite likely that a whole segment will fail due to an interruption of the cable (as seen in the recent Hunga-Tonga Hunga-Ha'apai event). Additionally, it is quite likely that in spite of the excellent quality of cabled pressure sensors, the data will not be just failing, but will be perturbed by individual uncertainties. Both these cases may alter the results substantially and need to be tested.

3. A further sensitivity relates to the selection of input. It should be discussed if wave forms are the best choice and how sensitive this choice of input really is. The authors use a maximum vs. mean wave height measure for input. But would it be possible - for a fast first warning product - to use time of first arrival? How could the (discarded) first 3 minute signal - mainly containing seismic wave impressions - be used in combination with a possible seismic model of the source?

4. Finally, consideration of the output sensitivity should be added: is the result sensitive to the choice of the output location grid? How sensitive is the model to the implicitly involved accuracy of the bathymetry and topography data?

5. In my opinion a (at least short) discussion on the limitations of neural network approaches with respect to "the unexpected", i.e. events that are outside of the expected range of scenarios, should be added. This is a major critique in the natural hazards community, since experience tells us that most of the devastating geohazards in the past decades gave rise to investigating unexpected magnitudes and features.

6. Some consideration on the usefulness of inundation data should be added as well. For example, in the early presentation of an analog forecasting system (using a simpler least squares type selection and combination algorithm) implemented in the InaTEWS system in Indonesia (Behrens et al., 2010) inundation computation is involved, but is not used for forecasting due to the involved uncertainty. Looking at figure 5, the local deviation of field measurements from computed inundation values is evident despite the statistical fit. So, what is the relevance of an inundation forecast, given this local uncertainty?

Answers in particular to the diverse uncertainty and sensitivity considerations should be given, before a publication in Nature can be considered. In its current state the manuscript to me appears already suitable for publication in more technical journals.

Minor comments

* Page 6 Figure 2b: The uplift and slip in these figures is hard to see in the current figure. A

different coloring may help here. the array of microplates could be plotted in a lighter color to have it less dominant, whereas the coloring and linewidth of the contour lines could be more prominent.

* Page 8, lines 168 ff: I do not understand the reasoning behind the necessary number of scenarios. Isn't it more important to design a proper sampling strategy to cover the whole parameter space, rather than the exact number of scenarios? If the scenarios are not chosen appropriately even a large number may not be suitable for the question, or am I missing something?

* Page 10, lines 210 ff: I think - regarding the major concern mentioned above - it is overly optimistic to diagnose excellent fit here. Yes the fit of the model result with a synthetic neural network based interpolation is excellent, but this is not relevant. I would suggest appropriate wording in this section.

* Page 14, lines 292 ff: You mention the (output) inundation grids, but you never define this properly. How did you chose this grid? What is a design criterion for this grid? what is the grid spacing and accuracy?

* Page 15, line 305: It would be interesting to see not only the absolute prediction uncertainty, but also the relative one. If the absolute uncertainty is 1.5 m, but this occurs in areas of 10 m flow depth, then this is less severe as a 30 cm uncertainty in an area of 1 m flow depth!

* Page 20, lines 428 ff: The use of Green's functions for wave form generation is somewhat unclear to me. Does this mean that the wave forms are then independent of the scenarios you use for training the neural network? or do you take the wave forms out of a selected number of scenarios in the training set? Please describe this in more detail.

* Page 23, lines 500/501: Please check the formulas again. I do not understand them. While you introduce variables K , P , and O , they show P_i and S_i , where is the S coming from and where did you use O ? Furthermore, the formula for κ appears flawed, since the factor $1/N$ is squared in the second term but not so in the first term.

Reference

J. BEHRENS, A. ANDROSOV, A. Y. BABEYKO, S. HARIG, F. KLASCHKA, L. MENTRUP (2010): A new multi-sensor approach to simulation assisted tsunami early warning, Nat. Hazards Earth Syst. Sci., 10:1085-1100, DOI:10.5194/nhess-10-1085-2010.

Reviewer #1

This paper proposed a machine learning method to forecast tsunami inundation (height and coverage) based on real-time information from offshore tsunami observing stations. The proposed method has the main advantage of estimating tsunami inundation without knowing the earthquake source model or initial tsunami source. Recently, machine learning techniques have been widely used in various research fields appealing to the remarkable ability to solve complex problems. The application of machine learning is not zero, but it is not so progressed in the tsunami research fields. In addition, predicting tsunamis in real-time is a great issue in tsunami disaster mitigation. Accordingly, the theme of the paper is very interesting. The paper is also well-written and structured; therefore, I recommend this paper for publication in Nature Communications after the authors addressed the following comments and questions:

Thank you very much for your efforts and time to comment on our work. We have carefully considered all comments as detailed below.

1. *P9, L175. The authors mentioned that the number of testing scenarios is 420. If 20 hypothetical megathrust earthquakes are generated from each magnitude (12 magnitudes), the total number of testing scenarios is 240?*

Response:

Thank you for noticing the error. The correct value is 40 scenarios for each magnitude, thus the total number of scenarios for the test set is $12 \times 40 = 480$ (not 420). We have corrected the typo in the referred line as follows:

“For the test set, we generate 40 hypothetical megathrust earthquakes for each magnitude (12 magnitudes) resulting in 480 scenarios.”

2. *Why do the authors only consider 33 fault models for the outer-rise earthquakes? Why don't the outer-rise scenarios be defined in the same manner as megathrust scenarios? See <https://doi.org/10.1002/2014JB010958> and [https://doi.org/10.1007/s00024-019-02364-](https://doi.org/10.1007/s00024-019-02364-4)*

4. Response:

Response to reviewer

The uncertainty of outer-rise fault parameters (particularly location or hypocenter) is significantly higher than that of the megathrust. To minimize subjectivity in defining the outer-rise faults, we used 33 models mapped by previous study (Baba et al., 2020^{#22}) that are based on marine survey and seismic monitoring results (e.g., Kodaira et al., 2017^{#23}; Boston et al., 2014^{#24}; Obana et al., 2019^{#25}; Fujie et al., 2016^{#40}). We refrain from partitioning the identified outer-rise faults into subfaults as Baba et al. (2020) suggested that the contribution of heterogeneous slips to the variability of tsunami heights is only 10–15%—much smaller than the megathrust earthquakes of similar magnitudes. To clarify this we added a sentence in paragraph 2 of the “Tsunami source scenario” section as follows:

“The coseismic displacements are calculated similarly to the megathrust earthquakes but with a uniform slip assumption. This is because the contribution of heterogeneous slips on the identified outer-rise faults to the tsunami variability is only 10-15%²².”

Both of the referred studies have been cited in the initial manuscript. The first one by Gusman et al. (2014^{#11}, <https://doi.org/10.1002/2014JB010958>) did not incorporate outer-rise faults, but discussed the potential of such faults to improve their proposed method. The second study by Fauzi and Mizutani (2019^{#13}, <https://doi.org/10.1007/s00024-019-02364-4>) included hypothetical outer-rise faults based only on the seismicity depth estimate, while the other fault parameters were speculatively determined. Note that both studies assumed a uniform slip distribution.

3. *The input of the machine learning model for the prediction is the maximum and mean tsunami amplitude, and eventually, the maximum value provides better results (for short time windows). Why do the authors use those values, not a whole part of waveforms? In this manner, I expect the prediction results to be much better.*

Response:

One of the advantages of our proposed method is simplicity. This is achievable with the dense and vast coverage of S-net. Our results demonstrate that with such a simple input structure we can produce comparable results to the physics-based model. Adding more information in the input (e.g., arrival times or waveforms) can also lead the training process complication, which

Response to reviewer

we believe is more suitable for smaller number of observation points as discussed in paragraph 4 of the Discussion. Further research is needed to find a suitable method to use waveforms in machine learning for tsunami inundation prediction application.

4. P3, line 65, typo “uncerteinties”? .

Response:

Corrected.

Reviewer #2

The manuscript “Machine learning-based tsunami inundation prediction derived from offshore observations” describes a AI-based approach to emulate tsunami simulations of inundation over large target areas based on offshore observations. I think that the presented research presents interesting results. However, I have some concerns that I describe in the followings, and that I think the authors can fully address in a new version of the manuscript.

Thank you very much for your efforts and time to comment on our work. We have carefully considered all comments as detailed below. The suggested modification of statistics to evaluate the results has been addressed and changes are made throughout the text and related Figures.

My main concern is about the selected procedure to demonstrate the goodness of the results. The adopted procedure is not very convincing to me, and I think that more must be done. In particular, the authors use, as main metric to evaluate results (reported also in the abstract, line 27), the “root mean square error” for several tsunami intensity measures (introduced in line 187). To my understanding, this is the root mean square of the residuals between the physics-based and the machine-learning-based tsunami simulations, averaged over all target points and all considered scenarios (please, see minor points regarding its definition). My fear is that this value may be strongly biased toward small values by a large number of target points (e.g. all the ones far from the coast) for which predicted and reference values are both practically equal to 0 (no tsunami). All target points far from the coast are in this situation, at least for all smaller magnitudes. This may introduce a large number of zeros into the evaluation of the root mean square that decreases a lot its final value. In practice, if this is the case, the root mean square error can be made as small as we wish simply by adding more points inland, where no tsunami will occur. This is somehow explicit in figure 6, where all points far from the coasts show values close to 0, while all points close to the coast have larger values. This means that the reported good numbers may be dominated by forecasts in areas in which the tsunami does not occur at all (or it is negligible), and small average residuals may be obtained almost independently from how good the model works where the tsunami actually occurs. Thus, I think that it could be useful:

Response to reviewer

- To focus the comparison on the areas in which the tsunami is expected, considering much smaller areas for smaller magnitudes (defined, for example, using as reference the physics-based simulations, and considering only points where the tsunami exceeds a lower threshold for inundation depth, e.g. 0.1 m).

Response:

We newly implemented a 0.2 m threshold (taken from the reference) for evaluating the flow depths, and the results are shown in Fig 4. Additionally, we also introduced three flow depth categories ($0.2 \text{ m} < h < 1 \text{ m}$, $1 \text{ m} < h < 3 \text{ m}$, and $h > 3 \text{ m}$, following tsunami warning level category of Japan Meteorological Agency) when evaluating the prediction of some selected samples on the test set (Supplementary Fig. 3-6) and real events (Fig. 7, and Supplementary Fig. 9-10).

- To propose a comparison also using relative differences, instead of absolute differences. Indeed, 1 m of misfit is maybe acceptable if 10 m are expected, while it may be not acceptable if 0.5 m is expected.

Response:

We introduced an additional statistical measure of accuracy in percentage from the K value (Equation 3) which was originally used to represent the relative differences (see Fig. 4a). We also reported a standard deviation of relative misfit (Fig. 5c) to improve the result presentation previously depicted in Fig. 6 of the initial manuscript.

- To report also the average of residuals (and not only the root mean square), to evaluate also the existence of potential biases in the predictions. To this end, also the statistics of the Aida's number (comparing physics-based and machine-learning-based results) could be useful.

Response:

We agree with the reviewer's comment on the drawback of the root mean square error (RMSE). Therefore, we replaced the RMSE with Aida's number including the additional measure of accuracy in percentage mentioned above. The results of this new evaluation are shown in Fig. 4a. We also use the Aida's number to evaluate the predictions of some selected samples and real events relative to the physics-based models.

Response to reviewer

- Given the importance of inundation areas, I suggest to introduce also some quantitative test about the correspondence of modelled inundation areas. In line 212, a “nearly perfect agreement” is stated for inundation areas, but no quantitative discussions about this quantity has been presented.

Response:

We added calculations of the inundation area (in km²) for the specified ranges of flow depth. The results are shown in all plots of inundation map (Fig. 7, Supplementary Fig. 9-10, and Supplementary Fig. 3-6).

To improve testing, it may be also useful to introduce some quantitative statistical test to rule out the existence of significant biases, for example demonstrating that the average misfit in the areas with significant tsunami is compatible with 0 (see, for example, the tests proposed in Davies 2019 - *Geophys. J. Int* - or in Selva et al. 2021 - *Nat. Comm.*), potentially keeping separated the scenarios for the different magnitudes.

Response:

We plotted a histogram (Fig. 4b) of misfits on the test set, following Selva et al. (2021), which suggests that the predictions are unbiased as the median is centered around zero. We also included a goodness-of-fit metric G (Equation 4) modified from Davies (2019) and plotted the results for each considered magnitude in Fig. 4c. Statistics on the evaluation result using the goodness-of-fit metric shown in the box plot are reported in Supplementary Table 1.

A second important concern is about the reported discussion about the “prediction uncertainty”. To my understanding, what the authors call “uncertainty” or “prediction uncertainty” (see, for example, line 344 or Fig. 6) is the average difference between physics-based simulations and machine-learning-based simulations. In my opinion, this is not the uncertainty on predictions, but the misfit between the two models. Prediction uncertainty is instead about the difference between models and the actual tsunami, and should include both input and modelling uncertainty for each individual tsunami. Thus, the discussed uncertainty is only a part of the total prediction uncertainty.

Response:

Response to reviewer

We agree with the reviewer. However, since actual tsunami data is limited, the only way to evaluate prediction results on the test set is by comparing them with the well-established physics-based model. In the revised manuscript, the referred section is placed before “Application to real events”. We changed the section name to “Spatial variability of misfits”. In the new plot of Fig 5, we also included a mean flow depth map (Fig. 5a) and a standard deviation of relative misfits (Fig. 5c) to see the relationship between large misfits with large expected tsunamis.

More specifically, most of the tests are made comparing physics-based and machine-learning tsunami models, and both training and testing datasets are produced using exactly the same tsunami model (same source, from SLAB2; same tsunami simulation scheme, etc., please see also minor comments about this). The limited comparison with real data demonstrate that physics-based and machine-learning methods perform similarly in predicting real data (lines 260-263, 285-289), but none of them is a “perfect prediction” (whatever this may mean). To well reproduce physics-based models is indeed an important results, but it is not an evaluation of the prediction uncertainty.

Response:

This actually has been addressed in the above comment, but we want to stress that we have changed the narrative on the prediction uncertainty and replaced some inaccurate qualitative assessments with the results of quantitative evaluations.

Therefore, on the one side, I suggest to use expressions like “modelling misfit” (or something similar) instead of “uncertainty” or “prediction uncertainty” throughout the manuscript. On the other side, I suggest to actually discuss the problem of prediction uncertainty, at least in the Discussion, as this is an important issue for the applicability and thus the potential impact of the proposed method. Machine-learning-based simulations are indeed very quick, and this may give the opportunity in the future to actually quantify prediction uncertainty using ensemble of different simulations (like proposed, for example, in Selva et al. 2021 - Nat. Comm. - or Giles et al, 2021 - Front. Earth Sci), quantifying the variability of the deterministic predictions over similar initial conditions or alternative modelling frameworks. This is somehow suggested in lines 367-369, but it is not properly discussed in the present version of the manuscript.

Response:

Response to reviewer

We changed the content of the previous “Prediction uncertainty” section and renamed it to “Prediction uncertainty relative to observation errors”. We defined the uncertainty as the prediction variability or sensitivity relative to inputs/observations, which is also suggested by reviewer #3. We added random perturbations to the input and reported the statistics of ensemble predictions in Fig. 8. However, this too does not represent the total prediction uncertainty as pointed out by the reviewer. Therefore, we added a paragraph in the Discussion section (paragraph 5) to explain the limitations of uncertainty quantification in this study.

A third important concern is about the differences between the presented procedure and the one recently published in Makinoshima et al. (2021 – Nat. Comm.), which share method (CNN), input data, and target area. I think that probably the authors have good points on their side, however I think that a deeper discussion is needed, to better demonstrate and highlight the effective novelties of this manuscript. For example, Makinoshima et al. state that their CNN simulations takes 0.004 s in a single CPU for a single target point, meaning that their method can be adopted in parallel for 100 or more points, still obtaining results in a fraction of second in a single CPU. In addition, differences between the adopted Neural Network configurations, or eventual differences in the pre-processing of the observation data (which are the same here and in Makinoshima et al.), should be further commented, in my opinion.

Response:

One of the main differences between our study and that of Makinoshima et al. (2021) lies in the model output structure. We believe that an inundation map is more practical in guiding evacuation route and straightforward to grasp by inexperienced users or the public in general than a time series. This is in line with many operational tsunami forecasting systems typically providing only maximum coastal tsunami heights. The target locations in our study area is > 200,000 points, if we apply Makinoshima et al. (2021) method, the 0.004 s for a location will lead to >800 sec (>13.3 min) processing time. Moreover, to store timeseries at >200,000 points for thousands of scenarios will require a huge disk space. We added the information on the target points in paragraph 5 of Introduction to highlight the importance of the proposed method focusing on the prediction of inundation map, which is also stressed in paragraph 6.

Other differences are indicated throughout the text:

Response to reviewer

- We used a more realistic source model from a wide range of potential megathrust and outer-rise earthquakes.
- We applied a statistical analysis to determine the efficient number of training set.
- The neural network architecture used in this study requires less computational specifications for the training as described in the Computing time section. Thus it can be easily reproduced by those who have limited access to expensive computational resources.
- Also, thanks to the reviewer, we have now included more rigorous evaluations to test the performance of the proposed method.

Finally, more attention should be devoted to discuss the potential practical applicability of the procedure in a real system, enriching the discussion in this sense. The arguments reported in the introduction (lines 97-101) are poorly discussed and not supported by specific literature, for example about how to inform in such a short time, or how to use such information during an ongoing evacuation. In addition, a link to the uncertainty on the forecast should be probably added, as already commented above.

Response:

We added a paragraph in the Introduction (paragraph 6) to discuss the practical applicability of the method and referred to previous related studies. Here, we also emphasized on the advantage and role of inundation map in guiding an evacuation process. Additionally, we also discussed about how to leverage an online tsunami modeling framework for an actual tsunami forecasting system.

“For near-field cases, spatial information on the maximum tsunami heights or flow depths is needed to inform the evacuation efforts that should start immediately following the detection of tsunamigenic earthquakes. A rapid inundation map could be used to enhance the available map for tsunami evacuation routes and safe zones. It will better inform the authorities involved (e.g., police, fire fighter, and operations manager) in executing tsunami evacuation plans. The efficacy of a real-time tsunami inundation map in guiding evacuees during a tsunami evacuation drill had been attested¹⁷. The inundation map is intuitively understandable to the public as it can be

Response to reviewer

overlaid with geographical maps and visualized via a common online map service, such as Google Maps¹⁷. Furthermore, cloud computing technology for online tsunami simulations has been proposed¹⁸. The on-demand computing environment is suitable and cost-effective for a real-time tsunami prediction occasionally accessed only during the event. For our case, the fully trained neural network parameters can be stored within a cloud instance and executed in real-time with the given inputs. The online modeling framework can also be configured to facilitate the warning dissemination to potential users.”

Other minor points are:

- Lines 103-104: please report the number (or the order of magnitude) of the target points. It would be interesting, for reproducibility of the results, to have the target grid as supplementary material.

Response:

The number of target locations is 214,356 points, which is reported in the Method (“Neural networks configuration” section). Now the number is also indicated in paragraph 5 of Introduction as mentioned above. We provided the coordinate of target points in the Supplementary file.

- Line 173: 33 is a very limited number to describe the potential variability of such sources. I suggest to comment this. I also suggest to report the list of the actual source parameters as supplementary material.

Response:

This is related to comment #2 by reviewer #1. We discussed the limitations of the outer-rise earthquake scenarios at several parts in the manuscript:

- “Tsunami source scenarios” section (paragraph 2)
“The tsunami source scenarios from outer-rise earthquakes are based on fault parameters provided by a previous study²². These faults were identified through marine seismic observations and surveys^{23–25}. The coseismic displacements are calculated similarly to the megathrust earthquakes but with a uniform slip assumption. This is because the contribution of heterogeneous slips on the identified outer-rise faults to the tsunami variability is only 10-15%²².”

Response to reviewer

- “Application to real events” section (paragraph 4)
“Ideally, more outer-rise faults should be incorporated into the data, albeit challenging as their identifications rely on the limited historical records or marine seismic observations and surveys^{22–25,38}”.

Baba et al. (2020^{#22}) provided the list of fault parameters used in this study. Therefore, instead of repeating the same information, we simply referred to the study in the “Tsunami source scenarios” section (paragraph 4) as follow

“A list of fault parameters for the outer-rise earthquakes used in this study can be found in Baba et al²².”

- Line 175: *the 480 scenarios used for testing are produced using the same simulation scheme used for training the model, both in terms of source and simulation scheme. This has to be better stressed, as it has consequences on how to interpret the results.*

Response:

We added the following sentences in paragraph 4 of the “Tsunami source scenarios” section to clarify the type of modeling framework for the training and test sets.

“Note that the training and test sets are based on the same simulation scheme for both tsunami generation and propagation. Thus, additional tests for different modeling frameworks are demonstrated through applications to three real events, which will be discussed later.”

Also in paragraph 1 of the “Application to real events” section:

“Methods and setups used to produce all the three source estimates of the real tsunami events are substantially different from those we apply to generate the data for training and test sets, particularly in terms of the fault geometry and discretization.”

- Line 187: *please define explicitly the “root mean square error”, for example with an equation in Method, section “Model Evaluation”; I struggled a bit to understand its actual definition.*

Response:

Response to reviewer

We have changed the statistical evaluation method as explained above, and root mean square error is no longer used. The new statistics are formulated in the Method “Model evaluations” section.

- Lines 211-212: I suggest avoiding words like “small” or “big”. Also, please use the same names in all the manuscript. Here, with “errors” you probably mean “root mean square error”, and it may be useful to introduce a specific symbol for it, as it is widely used in the manuscript.

Response:

Due to the new evaluation metrics, we rephrased all the related descriptions in the initial manuscript including that referred in this comment.

- Line 212: “nearly perfect” is too generic. Maybe inundation area should be properly tested, as suggested above.

Response:

We changed the wording and calculated the inundation area to quantitatively analyze the results as explained in the response to previous comments.

- Lines 235-236: consider to explain here why none of the events recorded by the network (which are mentioned before) cannot be used here.

Response:

We added the following sentence (paragraph 1 of “Application to real events” section).

“More recent tsunami events recorded by S-net²⁻⁴ were outside the training set distribution with no or minimum inundation traces, thus are not considered here.”

- Lines 242-249: are such sources modeled in the same 3D fault surface used before, or this is completely independent?

Response:

They are completely independent of the fault geometry used to generate the training set. In the revised manuscript, the effect of different source modeling is discussed in the “Application to real events” section (paragraph 3) as follows:

Response to reviewer

“However, the overall accuracy for the respective specified level of flow depth (Supplementary Fig. 9c) is inferior compared to the prediction for the 2011 Tohoku-oki event. This is partly because the source model of the 2011 Tohoku-oki event³⁵ (Supplementary Fig. 8a) was based on inverted surface displacements rather than fault slips. Therefore, the initial condition of the tsunami well matches the surface displacement profile from the more modern and complex plate geometry estimate²⁰ used to generate the training set (see Fig. 2 for examples). In contrast, the source model of the 1896 Meiji Sanriku event³³ (Supplementary Fig. 8b) was calculated from slips over a simplified fault geometry.”

- Line 247. please, be more quantitative and report more details about such substantial differences

Response:

We revise the sentence as follow.

“Methods and setups used to produce all the three source estimates of the real tsunami events are substantially different from those we apply to generate the data for training and test sets, particularly in terms of the fault geometry and discretization.”

- Lines 261-262 it may be useful to specify, here or in methods, that K close to 1 means no biases in the estimation, and that numbers $<$ or $>$ 1 correspond to over and under-estimations

Response:

We added the suggested descriptions in the Method section (“Model evaluations”) as follows:

“The prediction results are generally acceptable when the values of K and κ are within or close to the suggested criteria for satisfactory model performance, which are $0.8 < K < 1.2$ and $\kappa < 1.4$ ³⁸. An underestimation and overestimation of the observation are indicated by the K value larger and smaller than 1, respectively.”

- Fig.5. please, report also the difference between the physics-based and machine-learning-based results. In this way, it is almost impossible to evaluate. The comparison with data is not very effective with this representation. Maybe it can be enlarged and put in a separate panel, also reporting some histograms, etc

Response:

Response to reviewer

Figure 5 in the initial manuscript corresponds to Fig. 7 in the revised manuscript. We added a plot of the residual/misfit and statistical evaluation results. For comparisons with the observations, we keep the initial layout following the standard way to present them.

- Line 290: *I suggest to change name to this section. I suggest something related to the spatial variability of misfits*

Response:

This has been addressed in the previous comment. We renamed it to “Spatial variability of misfits and added a new section “Prediction uncertainty relative to observation errors”.

- Line 291-292: *with respect to what earlier is called “root mean square error”, the only difference here, is that you do not average in space, correct? If so, please specify*

Response:

As the referred content has been changed, this comment is no longer relevant.

- Line 302: *the mean is probably biased by all points with 0 misfit. Maybe, it is more appropriate to report some percentile (e.g., 90th percentile).*

Response:

The comment referred to Fig. 6 in the initial manuscript which has been modified in the revised manuscript (Fig. 5). We applied a different approach to analyze the misfit as explained in the “Spatial variability of misfits” section. The percentiles are now reported in Fig. 8, but for a slightly different purpose.

- Line 325: *“decent” is not very quantitative*

Response:

We removed the referred word.

- Lines 356-358: *see comment to lines 235-236*

Response:

We slightly modified the sentence as follows:

Response to reviewer

“However, the lack of real significant or large tsunami records presently registered at S-net renders further evaluations against real tsunami data a seemingly impossible task at the moment.”

- In figures 3-6 of the supplementary material, please add a plot of the misfit, in order to better appreciated differences

Response:

We added plot of misfits including statistical evaluation results as suggested.

- Lines 494-505: please, check this part, as it seems to me that there is an error here, between predicted P and observed (O) values. Note that O is not present in the equation.

Response:

Thank you for noticing the error. We have corrected the equations.

Reviewer #3

The manuscript of Mulia et al. describes the design and preliminary test of a tsunami forecasting system based on a neural network approach. The tensor flow based neural network is trained with approx. 3100 synthetic tsunami scenarios and is then tested by a number of simulated events - partly informed by field data - not contained in the training set. The results are compared with simulation results obtained by a physics-based forward simulation and the authors find that their neural network is 99% more computationally efficient, while producing acceptable deviations in the inundation flow depth. In this respect the findings are relevant and timely, since a number of articles in the past few years developed similar important progress in near field tsunami forecasting.

I have a few fundamental questions that I'd like to see addressed before I feel the manuscript acceptable for publication in Nature:

Thank you very much for your efforts and time to comment on our work. We have carefully considered all comments as detailed below.

1. The results or quality metrics of the forecasts are basically all synthetic. While the ground truth is based on hindcasts of real events, the input data for the neural network are all synthetic. Additionally, the error between the physical and the learned results are also reported in terms of comparing synthetic values - in weather forecasting this would be called a re-analysis. While this is an important first step towards a real forecasting system, it is nothing that marks a breakthrough and in particular is still prone to large uncertainty.

Response:

We agree with the reviewer on the limited test using real data. As indicated in paragraph 2 of Discussion, presently, there is no large tsunami record at S-net to test our method. But, we cannot wait until the disaster occurs to report our current findings. We have now added more evaluation procedures and uncertainty quantification to improve the quality of the paper. In the present work, we tried to avoid the complications associated with the tsunami source estimate, which is attainable with the use of a large-scale observing system such as S-net. The ability to predict the tsunami inundation without a source model is one of the main features of our

Response to reviewer

proposed method. Also, a single predictive model for a considerably wide area ($> 130 \text{ km}^2$) that can be executed within a fraction of a second can be a significant contribution to the disaster prevention efforts.

2. As a follow-up to my first point, I think the sensitivity analysis of the newly designed forecasting approach does not cover relevant possibilities. While the authors do consider the failure of individual inputs (basically, failing random individual sensors), more relevant realistic scenarios are not considered. Due to the design of the S-net sensor network in serial lines of sensors, in case of an earthquake rupture it is quite likely that a whole segment will fail due to an interruption of the cable (as seen in the recent Hunga-Tonga Hunga-Ha'apai event). Additionally, it is quite likely that in spite of the excellent quality of cabled pressure sensors, the data will not be just failing, but will be perturbed by individual uncertainties. Both these cases may alter the results substantially and need to be tested.

Response:

We included an additional scenario considering the failure of entire stations on a single S-net segment. The S-net segments are now depicted in Fig. 1 and the prediction accuracy corresponding to the segment failure is plotted in Fig. 6b. Fig 6.a shows the prediction accuracy corresponding to the failure of 93% (140 out of 150) of the sensors with random configurations. This scenario can also represent failures (partly or entirely) of some network segments.

In the revised manuscript, we changed the content of Prediction uncertainty section (also related to comment from reviewer #2). The uncertainty of predictions relative to the perturbed input at individual stations is now considered using the 2011 Tohoku-oki event as an example. The results from 100 ensemble predictions are shown in Fig. 8.

3. A further sensitivity relates to the selection of input. It should be discussed if wave forms are the best choice and how sensitive this choice of input really is. The authors use a maximum vs. mean wave height measure for input. But would it be possible - for a fast first warning product - to use time of first arrival? How could the (discarded) first 3 minute signal - mainly containing seismic wave impressions - be used in combination with a possible seismic model of the source?

Response:

Response to reviewer

Thank you for suggesting an interesting approach to use tsunami arrival times as input. As described in Introduction (paragraph 4) the neural networks prediction is based on statistical correlations between input and output irrespective of their physical relationships. Therefore, any give input can be correlated with its corresponding output, as long as the uniqueness of each sample/scenario is guaranteed. We agree that arrival time information will be useful to provide a fast warning product. In the revised manuscript, we added sentences and referred to a study by Behrens et al. (2010^{#54}), in paragraph 4 of Discussion as follows:

“Another type of input that can be extracted from tsunami data is the arrival time. Although it is not considered in this study, such information will be useful to provide a fast first warning product. The contribution of tsunami arrival times in reducing the prediction uncertainty has been reported in a previous study⁵⁴.”

Omitting the first 3 min of data is necessary because the long-period seismic signal typically occurs within this time frame shares similar spectral component with the tsunami. Therefore, as discussed in Saito and Kubota (2020^{#67}), it is difficult to remove such a seismic interference using a common low-pass filter, especially in real-time given the limited length of data. In the revised manuscript we also referred to a previous study by Inazu et al. (2016^{#68})—which also applied the same approach—in the “Input types and variability” section of Method as follows:

“Similar to a previous study⁶⁸, to avoid using corrupted data, we omit the first three minutes of pressure waveforms, during which the seismic wave disturbance predominantly occurs.”

4. Finally, consideration of the output sensitivity should be added: is the result sensitive to the choice of the output location grid? How sensitive is the model to the implicitly involved accuracy of the bathymetry and topography data?

Response:

We added a new section “Spatial variability of misfits”. This section discussed how the prediction skill varies with different coastal settings based on the present bathymetry and topography data (~30 m grid size). This variability is represented in terms standard deviations of misfit and relative misfit over the 480 scenarios on the test set (Fig. 5).

Response to reviewer

“The spatial variability of misfits and relative misfits on the test set can be represented by its standard deviation over all samples (Fig. 5b, c). Ideally, the standard deviation value should be close to zero, which indicates the prediction skill consistency at the same location in various earthquake scenarios. However, this is not always the case because of the high degree of nonlinearity of inundation flow depths, particularly over complex regions. Higher misfits are apparent near shorelines with complex coastal geometry and steep cliff typically characterizing the southern Sanriku coasts²⁹ (Fig. 5b). While the large standard deviation of misfits occurs sporadically in relatively small areas, clustered large values of more than 1.5 m are located at Ryori Bay. Yamanaka et al.³⁰ suggested that the tsunami height inside the Ryori bay is most likely amplified by the bay resonance. In our case, the expected flow depths in this area are also high, as shown by the mean flow depth of the target of up to 10 m (Fig. 5a), thus resulting in negligible relative misfits (Fig. 5c). This spatial variability of misfits can be thought of as one of the prediction uncertainty sources attributed to the model when approximating local effects from intricate coastal settings.”

5. In my opinion a (at least short) discussion on the limitations of neural network approaches with respect to "the unexpected", i.e. events that are outside of the expected range of scenarios, should be added. This is a major critique in the natural hazards community, since experience tells us that most of the devastating geohazards in the past decades gave rise to investigating unexpected magnitudes and features.

Response:

We modified paragraph 3 of Discussion to address the reviewer’s comment as follows:

“Another limitation of our method is related to spurious predictions when many S-net stations malfunction leading to inconsistent inputs with the training set distribution, as demonstrated by our synthetic experiments (Fig. 6). Furthermore, although we have considered a wide range of tsunami scenarios, we cannot rule out the possibility of unprecedented events that are outside the

expected range. Consequently, expert judgments for data quality assurance are needed to prevent unexpected outcomes. An automated data quality control can also be implemented, which is similar to that used in the seismic¹ and atmospheric⁴⁷ observations.”

6. Some consideration on the usefulness of inundation data should be added as well. For example, in the early presentation of an analog forecasting system (using a simpler least squares type selection and combination algorithm) implemented in the InaTEWS system in Indonesia (Behrens et al., 2010) inundation computation is involved, but is not used for forecasting due to the involved uncertainty. Looking at figure 5, the local deviation of field measurements from computed inundation values is evident despite the statistical fit. So, what is the relevance of an inundation forecast, given this local uncertainty?

Response:

In the Introduction (paragraph 6), we discussed the role of inundation prediction in guiding evacuation processes referring to a previous study by Gusman et al. (2015^{#17}). We also discussed the potential of an online modeling framework proposed by Behrens et al. (2022^{#18}) in relation with the practical applicability of our proposed method.

It is true that there are some considerable deviations from the measurement at several locations, which is also apparent in the physics-based model result. Among other factors (e.g., measurement errors, model simplifications), the quality and spatial resolution of bathymetry and topography contribute to the said discrepancies. In this study, we used a 30-m grid size. If data with higher resolution (for example ~5 or 1 m) is available, we believe that the local uncertainty associated with bathymetry and topography can be reduced. The smaller grid size will substantially increase the computing time of the physics-based model, but not necessarily for the machine learning-based model. We added descriptions related to this in the “Computing time” section (paragraph 1) as follow.

“The remarkable computing speed will also facilitate further improvements of the prediction when a higher bathymetry and topography resolution is considered. A smaller grid size would improve the accuracy and suppress the uncertainty associated with bathymetry and topography data.”

Response to reviewer

Despite the said limitation, the predicted inundation map of the present result is an improvement to the existing tsunami forecasting systems typically provide only maximum tsunami heights along the shore.

Minor comments

** Page 6 Figure 2b: The uplift and slip in these figures is hard to see in the current figure. A different coloring may help here. the array of microplates could be plotted in a lighter color to have it less dominant, whereas the coloring and linewidth of the contour lines could be more prominent.*

Response:

We modified the figure as suggested.

** Page 8, lines 168 ff: I do not understand the reasoning behind the necessary number of scenarios. Isn't it more important to design a proper sampling strategy to cover the whole parameter space, rather than the exact number of scenarios? If the scenarios are not chosen appropriately even a large number may not be suitable for the question, or am I missing something?*

Response:

There are many ways to select samples that sufficiently cover the whole parameter space, and our approach is one of them. Here, the level of input (maximum tsunami height at S-net) variability is an indication of the parameter space. As explained in the Method (“Input types and variability” section), an appropriate number of scenarios—expected to represent the whole parameter space—for each considered earthquake magnitude is achieved when the variability of corresponding inputs reaches a stable state marked by the red dashed line in Fig. 3a. This analysis prevents the use of redundant scenarios that are no longer contributing to variability the training set distribution.

** Page 10, lines 210 ff: I think - regarding the major concern mentioned above - it is overly optimistic to diagnose excellent fit here. Yes the fit of the model result with a synthetic neural network based interpolation is excellent, but this is not relevant. I would suggest appropriate wording in this section.*

Response:

Response to reviewer

We updated the results with quantitative assessments (Supplementary Fig. 3-6) and changed the wording in paragraph 4 of “Prediction results on the test set” section as follows:

“Overall, the predictions exhibit good agreement with the targets, as indicated by statistical evaluation results shown in the respective figure.”

** Page 14, lines 292 ff: You mention the (output) inundation grids, but you never define this properly. How did you chose this grid? What is a design criterion for this grid? what is the grid spacing and accuracy?*

Response:

In the Method (“Physics-based tsunami simulation” section) we specified the grid sizes for the tsunami modeling including the availability of the 30 m grid topography data considered as the finest grid resolution in this study. In the revision, we also provided the inundation grid coordinates in the Supplementary file. Additionally, in the Method (“Neural network configuration” section) we defined the total number of inundation grids corresponding to the output nodes of the neural networks.

** Page 15, line 305: It would be interesting to see not only the absolute prediction uncertainty, but also the relative one. If the absolute uncertainty is 1.5 m, but this occurs in areas of 10 m flow depth, then this is less severe as a 30 cm uncertainty in an area of 1 m flow depth!*

Response:

We plotted the relative misfit as shown in Fig 5c under the “Spatial variability of misfits” section. The relative misfit is defined in the Method “Model evaluation” section.

** Page 20, lines 428 ff: The use of Green's functions for wave form generation is somewhat unclear to me. Does this mean that the wave forms are then independent of the scenarios you use for training the neural network? or do you take the wave forms out of a selected number of scenarios in the training set? Please describe this in more detail.*

Response:

The waveforms from the Green’s functions summation are independent of waveforms used for the training set. The waveforms from the Green’s functions are used only for defining the

Response to reviewer

appropriate number of scenarios as explained in the response to the previous comment. Once we obtained the number of scenarios that are expected to represent the whole parameter space, we run the tsunami simulations using the same slip models used during the Green's functions analysis. The waveforms at S-net and the corresponding inundations from these actual forward model simulations are then used as inputs and outputs/targets for the neural networks training and test sets.

We now stress this in paragraph 3 of the "Tsunami source scenarios" section to avoid misunderstanding as follows:

"To determine the appropriate number of scenarios for the megathrust earthquakes, we first apply a Green's functions summation technique^{26,27} described in the Method section. With this technique, we can efficiently generate tsunami waveforms at S-net stations from many earthquake slip scenarios before simulating the inundation. Therefore, this step is independent of the actual simulation of tsunami scenarios used for the training and testing sets."

** Page 23, lines 500/501: Please check the formulas again. I do not understand them. While you introduce variables K , P , and O , they show P_i and S_i , where is the S coming from and where did you use O ? Furthermore, the formula for κ appears flawed, since the factor $1/N$ is squared in the second term but not so in the first term.*

Response:

Thank you for noticing the error. We have corrected the typo and confirmed that the formula is consistent with the original paper by Aida (1978^{#75}).

REVIEWERS' COMMENTS

Reviewer #2 (Remarks to the Author):

This version has several very significant improvements with respect to the first manuscript. The authors included a more detailed analysis of the misfits, which allow a much more detailed comprehension of the results. I found particularly interesting the new analysis proposed to discuss the uncertainty due to measurement errors (Section "Prediction uncertainty relative to observation errors", lines 403-434 in the track-change version). I think that this analysis clearly demonstrates that this tool is perfectly complementary to probabilistic tsunami forecasting accounting for uncertainty, for which many simulations accounting for input errors and simulation uncertainties are required.

Overall, I think that the manuscript is in good shape and that it could be accepted after addressing a few minor points (see below: lines are always referred to the track-change version).

Jacopo Selva

Minor points:

- Lines 94-104: This discussion clearly improved. However, I feel that it is still missing the comparison between the arrival time of the tsunami in the near-field, and the first results available from this method, which are > 10/15/20 minutes for the different configurations. Notably, this quantification may be useful also after inundation occurred to inform the help on the ground.
- Line 235: it is hard here to say if the obtained median values of G are good or not. They are small, and this is good, but we do not know how much small they should be, I think. Then, I suggest to move above the sentence "Nonetheless, the present results are acceptable, with the overall median of G ranging between 0.004-0.086 for all magnitudes", linking this to the fact that 15-85 percentiles are < 1 m in Fig. 4b, and probably also higher percentiles (1-99) are smaller than 2 m, more than to the results of G.
- Lines 276-278: I do not understand this sentence. The spatial variability is a very input information, that can be used to evaluate the prediction uncertainty in the different locations. However, I do not think that the spatial variability can be considered a "source of prediction uncertainty".
- Lines 326-328: As acknowledged here, there are limits depending on minimum records at the S-net. Can be this translated in terms of magnitudes and source area?
- Line 350-351: this is very interesting. Can you discuss possible reasons for this?
- Line 403-434: I found very interesting the analysis of the uncertainty due to measurement errors (Section "Prediction uncertainty relative to observation errors", lines 403-434 in the tracked-change version). This definitely goes in the direction of demonstrating that this tool is complementary to probabilistic forecasting accounting for uncertainty, as commented in the Discussion (line 517-530). This example perfectly reaches this goal, so I suggest to state it explicitly.

Other suggestions / notes:

- Line 27: I don't know if "preclude" is the right word. Maybe "overtake", "eliminate"?
- Line 80: Multitude  large number?
- Line 81: maybe there is a missing "in" before citations 13, 14
- Line 146: why "particularly after 2011 megathrust earthquake"? this should be justified. All the examples reported afterwards, are pre 2011. This statement may also be deleted, as it is not very relevant in this context.
- Line 181: I would rather say that "about 3000 scenarios are required (following fig. 3, we use 3060)", or something like this. To state that you require exactly 3060 scenarios sounds strange to

me, at least.

- Line 183: the presence of the citation confused me (it looks like 33^{22} ...), but maybe in the edited version this will be solved.
- Line 212: I would suggest to add a name or a short description to the parameters that you are going to use in the followings, to specify in few words what the 3 equations evaluate before commenting Fig4 results.
- Line 220-226: I suggest to round all percents at the closest units (here and in the following)
- Line 224: it is hard at this point of the manuscript to judge why 92.6% (for maximum) is ok, and instead 82% (for mean) is too small. From Fig. 4a, I would say that mean and maximum work similarly, but maximum is preferred as it has (slightly) better results in 10 min and, maybe, also because it is simpler to evaluate.
- Line 235: probably the citation 27 is wrongly positioned
- Line 338: "would" or "can"?
- Line 345: "a" should be removed
- Line 403: may be better "due to" than "relative to"?
- Line 548: see comment for line 80

Reviewer #3 (Remarks to the Author):

This reviewer thanks the authors for a thorough and extensive revision, which has benefited the manuscript considerably. I agree with almost all changes and congratulate the authors for wise decisions taken.

With this, I can recommend the manuscript for publication.

I have only a few minor remarks that will not be of major influence to the quality, but may be considered for the final version:

1. Figure 5 is hardly visible, maybe zoom into areas of interest?
2. Figure 8 is the same, very hard to identify anything, maybe highlight areas?
3. Line 452-455: I am not sure if higher resolution bathymetry actually decreases uncertainty wrt bathymetry. Yes, the data may be more accurate, but now the model may not be adequate any longer!

(Line numbers refer to the document with color-indicated changes)

Reviewer #2

This version has several very significant improvements with respect to the first manuscript. The authors included a more detailed analysis of the misfits, which allow a much more detailed comprehension of the results. I found particularly interesting the new analysis proposed to discuss the uncertainty due to measurement errors (Section “Prediction uncertainty relative to observation errors”, lines 403-434 in the track-change version). I think that this analysis clearly demonstrates that this tool is perfectly complementary to probabilistic tsunami forecasting accounting for uncertainty, for which many simulations accounting for input errors and simulation uncertainties are required.

Overall, I think that the manuscript is in good shape and that it could be accepted after addressing a few minor points (see below: lines are always referred to the track-change version).

Jacopo Selva

Dear Dr. Selva. Thank you very much for taking the time to review the paper thoroughly. We can see the improvement of the current submission compared to the initial one, which is mainly based on your valuable comments and suggestions. The followings are our response to your additional minor comments.

1. *Lines 94-104: This discussion clearly improved. However, I feel that it is still missing the comparison between the arrival time of the tsunami in the near-field, and the first results available from this method, which are > 10/15/20 minutes for the different configurations. Notably, this quantification may be useful also after inundation occurred to inform the help on the ground.*

Response:

Currently, our method does not consider the arrival time as one of the predicted variables. We believe that, for near-field cases, the evacuation should be started immediately once the model predicts the inundation potential at the areas of interest. This is stated in the first sentence of the referred paragraph. However, arrival times at each of the inundated grid can also be modeled and

Response to reviewer

treated similarly to the flow depth using the same model setting and architecture, which will probably be considered in the future study.

2. *Line 235: it is hard here to say if the obtained median values of G are good or not. They are small, and this is good, but we do not know how much small they should be, I think. Then, I suggest to move above the sentence "Nonetheless, the present results are acceptable, with the overall median of G ranging between 0.004-0.086 for all magnitudes", linking this to the fact that 15-85 percentiles are < 1 m in Fig. 4b, and probably also higher percentiles (1-99) are smaller than 2 m, more than to the results of G .*

Response:

We have rephrased the sentences as follow:

"Nonetheless, the present results are generally acceptable considering the misfit between 15th and 85th percentiles is only < 1 m (Fig 4b). The overall median of G ranging between 0.004-0.086 for all magnitudes."

3. *Lines 276-278: I do not understand this sentence. The spatial variability is a very input information, that can be used to evaluate the prediction uncertainty in the different locations. However, I do not think that the spatial variability can be considered a "source of prediction uncertainty".*

Response:

We changed the sentence as follow:

"This spatial variability of misfits can be thought of as one of the model limitations when approximating local effects from intricate coastal settings."

4. *Lines 326-328: As acknowledged here, there are limits depending on minimum records at the S-net. Can be this translated in terms of magnitudes and source area?*

Response:

We designed our machine learning-based model to take inputs only from S-net data because of the relatively large number of stations. Using only magnitude and source area as the basis to generate inundation predictions can be challenging, although not impossible. One important

Response to reviewer

requirement for machine learning modeling is to ensure the sample uniqueness, which is difficult to achieve with limited number of inputs.

5. *Line 350-351: this is very interesting. Can you discuss possible reasons for this?*

Response:

It is difficult to trace the source of such a discrepancy of error between the small and large tsunamis. One possible explanation is that the neural networks are more heavily weighted toward large outputs because we did not normalize the value due to the use of a rectified linear unit (ReLU) activation function. However, this is still a speculation. Further investigations should be carried out to better understand the underlying causes.

6. *Line 403-434: I found very interesting the analysis of the uncertainty due to measurement errors (Section "Prediction uncertainty relative to observation errors", lines 403-434 in the tracked-change version). This definitely goes in the direction of demonstrating that this tool is complementary to probabilistic forecasting accounting for uncertainty, as commented in the Discussion (line 517-530). This example perfectly reaches this goal, so I suggest to state it explicitly.*

Response:

We appreciate the reviewer's comment on the new analysis about the uncertainty quantification. We included additional sentence in the abstract to state and highlight the uncertainty analysis achieved in this study more clearly.

"The proposed machine learning-based model can achieve comparable accuracy to the physics-based model with ~99% computational cost reduction, thus facilitates a rapid prediction and an efficient uncertainty quantification."

Other suggestions/notes:

- *Line 27: I don't know if "preclude" is the right word. Maybe "overtake", "eliminate"?*
- *Line 80: Multitude  large number?*
- *Line 81: maybe there is a missing "in" before citations 13, 14*

Response:

All the above have been corrected.

Response to reviewer

• *Line 146: why "particularly after 2011 megathrust earthquake"? this should be justified. All the examples reported afterwards, are pre 2011. This statement may also be deleted, as it is not very relevant in this context.*

Response:

We deleted the referred parts:

"The other important tsunami source is the outer-rise earthquake (Figure 2a)."

• *Line 181: I would rather say that "about 3000 scenarios are required (following fig. 3, we use 3060)", or something like this. To state that you require exactly 3060 scenarios sounds strange to me, at least.*

Response:

We agreed, but we still need to state the exact number of scenarios to be consistent with the subsequent sentences on the addition of outer-rise events. The sentences have been rephrased as follow:

"We then apply a linear interpolation to the remaining magnitudes illustrated by the red dashed line in Fig. 3a, resulting in 3060 scenarios. This implies that about 3000 plate interface earthquake scenarios are needed for the training set."

• *Line 183: the presence of the citation confused me (it looks like 33^{22} ...), but maybe in the edited version this will be solved.*

Response:

We removed the citation to avoid confusion. The same citation can be found in the subsequent sentence.

• *Line 212: I would suggest to add a name or a short description to the parameters that you are going to use in the followings, to specify in few words what the 3 equations evaluate before commenting Fig4 results.*

Response:

We added the evaluated property in the new sentence as follow:

Response to reviewer

“To evaluate the results which are predicted flow depths at inundated grids, we apply Equations 1-3 to all locations of flow depths ≥ 0.2 m and over all samples on the test set simultaneously.”

- *Line 220-226: I suggest to round all percents at the closest units (here and in the following).*

Response:

We kept the current format to be consistent with those indicated in Figs. 4a, 7c, and other Figures in the supplementary. Changing this would require to update all the figures.

- *Line 224: it is hard at this point of the manuscript to judge why 92.6% (for maximum) is ok, and instead 82% (for mean) is too small. From Fig. 4a, I would say that mean and maximum work similarly, but maximum is preferred as it has (slightly) better results in 10 min and, maybe, also because it is simpler to evaluate.*

Response:

We rephrased the sentences as follow:

“These results indicate that both types of input exhibit comparable accuracy, but the input of maximum amplitude outperforms the mean amplitude particularly at a relatively short time window. For conciseness, hereafter, our discussion refers only to the results using maximum tsunami amplitudes as input within the 20 min prediction window.”

- *Line 235: probably the citation 27 is wrongly positioned*
- *Line 338: "would" or "can"?*
- *Line 345: "a" should be removed*
- *Line 403: may be better "due to" than "relative to"?*
- *Line 548: see comment for line 80*

Response:

All the above have been corrected.

Response to reviewer

Reviewer #3

This reviewer thanks the authors for a thorough and extensive revision, which has benefited the manuscript considerably. I agree with almost all changes and congratulate the authors for wise decisions taken.

With this, I can recommend the manuscript for publication.

I have only a few minor remarks that will not be of major influence to the quality, but may be considered for the final version:

Thank you very much for the comments and suggestions in the first round which greatly helped to improve the quality of our manuscript. Also, thank you for agreeing with the first revision. The followings are our response to your additional minor comments.

- 1. Figure 5 is hardly visible, maybe zoom into areas of interest?*
- 2. Figure 8 is the same, very hard to identify anything, maybe highlight areas?*

Response:

We updated Figure 5 as suggested but kept Figure 8 as the initial plot because we do not refer to particular areas when discussing the result.

- 3. Line 452-455: I am not sure if higher resolution bathymetry actually decreases uncertainty wrt bathymetry. Yes, the data may be more accurate, but now the model may not be adequate any longer!*

Response:

We removed the referred sentence to avoid making an unjustified statement.